# On the Impact of Performative Risk Minimization for Binary Random Variables

**Nikita Tsoy** [1]   **Ivan Kirev** [1]   **Negin Rahimi** [1]   **Nikola Konstantinov** [1]

## Abstract

Performativity, the phenomenon where outcomes are influenced by predictions, is particularly prevalent in social contexts where individuals strategically respond to a deployed model. In order to preserve the high accuracy of machine learning models under distribution shifts caused by performativity, Perdomo et al. (2020) introduced the concept of performative risk minimization (PRM). While this framework ensures model accuracy, it overlooks the impact of the PRM on the underlying distributions and the predictions of the model. In this paper, we initiate the analysis of the impact of PRM, by studying performativity for a sequential performative risk minimization problem with binary random variables and linear performative shifts. We formulate two natural measures of impact. In the case of full information, where the distribution dynamics are known, we derive explicit formulas for the PRM solution and our impact measures. In the case of partial information, we provide performative-aware statistical estimators, as well as simulations. Our analysis contrasts PRM to alternatives that do not model data shift and indicates that PRM can have amplified side effects compared to such methods.

## 1. Introduction

Predictions can significantly influence everyday life (Rodamar, 2018), an effect known as performativity. For instance, traffic predictions can alter people's daily routes, crime predictions can affect police resource allocation, and stock price predictions can steer traders' decisions. These changes can lead to shifts in the underlying data distribution, making the original predictions less accurate.

To capture these effects, Perdomo et al. (2020) introduced the concept of *performative prediction*. In this framework,

a deployed model $\theta$ induces a data distribution $D(\theta)$, which gives rise to a new learning objective, the *performative risk* $\mathbb{E}_{z \sim D(\theta)}(\ell(\theta, z))$, with $\ell$ being a loss function and $z$ being a sample from the model-induced distribution. Much progress has been achieved in performative risk minimization (PRM), i.e., finding performatively-optimal points (see Hardt & Mendler-Dünner, 2023, for a recent survey).

While PRM is preferable to standard risk minimization (RM) for the sake of test-time accuracy, the broader impact of PRM remain elusive. In particular, using PRM instead of RM leads to different predictions deployed by the learner and also changes the evolution of the data distribution, and these effects are compounded when deploying multiple models over time. This limited understanding of the impact of PRM on the predictions and distribution may be partially due to the mathematical challenges arizing from analyzing the long-term dynamics of the learned models, in the presense of intricate dependencies of the data distribution on all previous models.

**Contributions**  In this work, we initiate the analysis of the broader impact of PRM, by studying a sequential performative mean estimation problem for binary variables, in the presence of linear performative distribution shifts. The simplicity of the learning setup enables us to derive the long-term dynamics of PRM, despite the complicated downstream impact of each deployed prediction on the future data distributions. This in turn allows us to quantify the evolution of the predictions and the data distribution.

Within this model, we formulate two measures of impact. The first measure concerns the model predictions and corresponds to the usual statistical notion of a bias of an estimator. The second measure quantifies the shift in the mean of the binary random variable, relative to the mean in the case of lack of performative effects, and thus allows us to understand the evolution of the data distribution under PRM.

We analyze PRM and the two measures in a one-period (single model deployment) and an infinite horizon (sequential model deployment) setting. In each case, we first study a full information setting, where all problem parameters (e.g. strength of performativity and initial distribution) are known to the model provider, in order to isolate the effects of performativity from exploration. We then analyze performativity

[1]INSAIT, Sofia University "St. Kliment Ohridski". Correspondence to: Nikita Tsoy <nikita.tsoy@insait.ai>.

*Proceedings of the $42^{nd}$ International Conference on Machine Learning*, Vancouver, Canada. PMLR 267, 2025. Copyright 2025 by the author(s).

and exploration jointly via theory and simulations.

Our results indicate that, compared to RM, PRM may select more biased estimators and/or ones that shift the mean to extreme values. This happens in particular because minimizing the PRM loss suggests trading-off the usual mean squared error (MSE) for reduced aleatoric uncertainty in the future data distribution. Such effects occur when the distribution responds positively to model predictions or when the distribution responds negatively, but the model is updated rapidly and the performativity is high.

Finally, we use two example scenarios to interpret our measures and technical results in a social context.[1]

## 2. Related Work

**Performative Prediction**    In machine learning, performativity is often studied within the framework of *performative prediction*, where the goal is to find a model with good performance on the distribution that it induces. The setting was introduced by Perdomo et al. (2020) and was inspired by works on strategic classification (Hardt et al., 2016; Dalvi et al., 2004). Numerous works study methods for finding performatively optimal/stable models (Mendler-Dünner et al., 2020; Miller et al., 2021; Jagadeesan et al., 2022; Izzo et al., 2022; Ray et al., 2022b; Lin & Zrnic, 2024), see Hardt & Mendler-Dünner (2023) for a recent overview. Brown et al. (2022); Ray et al. (2022a); Mandal et al. (2023) extend this framework to stateful environments, where previous model deployments impact the data distribution at later stages.

In contrast to the works above, we focus on the impact of PRM on the data distribution and on the predictions made by these models. To our awareness, the only work that studies properties beyond performative loss in the context of performative prediction is that of Jin et al. (2024), who, however, focus on the fairness and polarization properties of PRM instead.

**Distribution Steering**    Our work analyzes the secondary effects of PRM on the distribution and outcomes, which were not intended by the model provider. Several related works could allow the model provider to encode penalties for these unintended changes into its optimization task. Kim & Perdomo (2023) investigate how to steer distributions towards a more desirable outcome by using omnipredictors. Similarly, Golowich et al. (2024) study the task of distribution steering in population dynamics context. However, these results do not inform about the unintended distribution changes due to PRM, which is the focus of our work.

**Long-Term Fairness**    The line of works on long-term fairness also studies the evolution of distribution in social contexts. Ensign et al. (2018b); Bechavod et al. (2019) focus on social feedback loops. Williams & Kolter (2019); Liu et al. (2020) propose models for performative responses motivated by their learning context. While these works model performativity, they focus on finding fair models. In contrast, we focus on performatively optimal algorithms and analyze their impact on the data distribution and predictions.

**Instances of Performativity**    Performativity arises in many social contexts. Economic agents respond to the actions of the government (Lucas, 1976). Performative policing affects the distribution of observed crime rates (Ensign et al., 2018a). Traffic predictions reroute drivers to new areas (Macfarlane, 2019; Cabannes, 2022). Recommendation systems affect the consumption of new content (Brown & Agarwal, 2022; Dean & Morgenstern, 2022). Since performativity is so widespread, it is important to study optimization formulations in such settings and the effects of performatively-optimal solutions on their environment.

## 3. Model

We now discuss the sequential performative prediction framework we study, the specific instance considered in our analysis and the impact metrics we focus on.

### 3.1. Performative Prediction Framework

**Optimization Problem**    This work analyzes how optimizing for performative accuracy influences the model provider actions and the underlying probability distribution. To answer this question, we consider the following sequential performative prediction problem inspired by Perdomo et al. (2020); Brown et al. (2022); Ray et al. (2022a). Denote by $\Theta$ the space of models, by $\theta_t \in \Theta$ the model parameters at time $t$, by $\mathcal{D}$ the space of all data distributions on a data space $\mathcal{Z}$, by $D_t \in \mathcal{D}$ the data distribution after the response to $\theta_{t-1}$, and by $\Psi \colon \mathcal{D} \times \Theta \to \mathcal{D}$ the model of performative response, such that $D_t = \Psi(D_{t-1}, \theta_{t-1})$. The model provider is interested in minimizing a discounted loss

$$\min_{(\theta_t)_{t=0}^{T-1}} \mathbb{E}_{(\theta_t)_{t=0}^{T-1}} \left( \sum_{t=0}^{T-1} \gamma^t \, \mathbb{E}_{\boldsymbol{z} \sim D_t^{\text{test}}}(\ell(\theta_t, \boldsymbol{z})) \right), \quad (1)$$

where $\ell$ is the loss function, $\gamma \in (0,1)$, $T \in \mathbb{N} \cup \{\infty\}$, and $\theta_t$ depends only on the information up to time $t$. We denote the solution to this problem by $(\theta_t^*)_{t=0}^{T-1}$ and refer to it as the *PRM path*. Conceptually, this path can be seen as the "performatively-optimal" sequence of models (Perdomo et al., 2020).

*Remark* 3.1. Notice that our problem falls within the framework of reinforcement learning under partial observability, where $\theta_t$ corresponds to action and $D_t$ corresponds to state.

---

[1] Please see the replication files for our paper at `https://github.com/insait-institute/performative-prediction-impact-replication`

**Test Distribution** The objective (1) depends on the model of test distributions $D_t^{\text{test}}$. In the standard performative setting (Perdomo et al., 2020), $D_t^{\text{test}} = D_{t+1}$, the model is tested in an environment adapted to it. This property holds when the speed of model deployment is slower than that of societal adaptation. Thus, we call this case the *slow deployment* case. For example, drug efficacy estimates can only be updated after time-consuming clinical trials.

We also consider the case of $D_t^{\text{test}} = D_t$, when the environment adapts to the predictions with delay. Such delays arise whenever models are updated frequently. Therefore, we call this case the *rapid deployment* case. For example, the predictions of road congestion can be updated "on the fly", so people may not be able to adapt to the latest predictions.

### 3.2. Instance of Performative Problem

**Distribution** We assume that $D_t$ describes binary random variables $z \sim D_t$ with mean $p_t$

$$z = \begin{cases} -1/2, & \text{w.p. } 1/2 - p_t, \\ 1/2, & \text{w.p. } 1/2 + p_t. \end{cases}$$

Note that $z$ is a Bernoulli random variable shifted by $1/2$ for mathematical convenience. For these variables, a positive outcome could mean that a drug is effective for treating a patient or that a certain route is free from traffic.

**Loss** At time $t$, the model provider deploys $\theta_t \in [-1/2, 1/2]$ to minimize mean squared error (MSE) $\ell(\theta_t, z) := (\theta_t - z)^2$. We denote the expected loss (w.r.t. all randomness) by

$$\texttt{loss}_t := \mathbb{E}(\mathbb{E}_{z \sim D_t^{\text{test}}}((\theta_t - z)^2)).$$

We denote the means produced by the PRM path $(\theta_t^*)_{t=0}^T$ by $(p_t^*)_{t=0}^T$. We also denote by $p_t^{\text{test}}$ the mean of the distribution $D_t^{\text{test}}$. Note that $p_t^{\text{test}}$ is equal to $p_{t+1}$ and $p_t$ in the slow and rapid deployment cases, respectively.

**Lemma 3.2** (Error-Uncertainty Tradeoff)**.** *The mean squared error of $\theta_t$ on $D_t^{test}$ is*

$$\mathbb{E}((\theta_t - z)^2 \,|\, \theta_t, p_t^{test}) = (\theta_t - p_t^{test})^2 + (1/4 - (p_t^{test})^2). \quad (2)$$

Here, the first term corresponds to the standard mean squared error (MSE). The second term corresponds to the aleatoric uncertainty of $D_t^{\text{test}}$ (note that such decompositions are valid for a big class of distributions, Gupta et al., 2022). Thus, under performativity, the model provider is also incentivized to decrease the environment uncertainty, while in the non-performative case they only minimize the MSE.

**Performative Response** Performativity manifests differently in different contexts. For example, route congestion

estimates might have *negative feedback* on the congestion: when the model predicts that one route is less busy than others, people might use it more. On the other hand, drug efficiency estimates might have *positive feedback* on the drug efficacy due to the well-known placebo effect.

We capture these effects using a linear response model

$$p_{t+1} := \alpha\theta_t + (1 - |\alpha|)s_{t+1}, \quad (3)$$

where $s_{t+1} := \lambda p_t + (1 - \lambda)\pi$, $\alpha \in (-1, 1)$, $\lambda \in [0, 1)$, and $\pi \in [-1/2, 1/2]$. Here, $s_{t+1}$ is the next period mean in the absence of performativity, $\alpha$ controls the strength and direction of performativity, $\lambda$ controls the friction in the distribution update, and $\pi$ is the equilibrium (long-term) mean in the absence of model influence. Positive $\alpha$ describes positive feedback situations. Negative $\alpha$ describes negative feedback situations. We also use the notation $\beta := (1 - |\alpha|)\lambda$, under which $p_{t+1} = \alpha\theta_t + \beta p_t + (1 - |\alpha| - \beta)\pi$.

**Limitations** This work considers a specific instance of our general performative framework to get a comprehensive theoretical description of the considered impact metrics (see Section 3.3). While the considered form of the problem limits generality, we believe that our analysis could be informative for real-world situations. Therefore, we discuss the potential applications and limitations of our analysis in detail in Section 6.

### 3.3. Measuring the impact of PRM

The distribution we consider is determined by its mean. This property allows us to formulate two natural "impact" metrics, *bias* and *mean shift*. Bias captures the *impact of PRM on the learner's predictions*, while mean shift describes the *impact of PRM on the underlying distribution*. We discuss the generalizations of these metrics to other learning tasks in Appendix A.1.

**Bias** Consider an arbitrary path (sequence of predictions) $(\theta_i)_{i=0}^T$. Inspired by the classic notion of bias, at each time $t$ we study the expected error in the estimate of the mean

$$\texttt{bias}_t := \mathbb{E}(\theta_t - p_t^{\text{test}}). \quad (4)$$

Intuitively, the bias captures how far (on average) are the predictions of the path from the true mean at a given time.

**Mean Shift** Here we compare the mean $p_t$ of the distribution under the path $(\theta_i)_{i=0}^T$ and corresponding mean in the absence of performativity $p_t^0$ (i.e., when $\alpha = 0$). Formally,

$$\texttt{shift}_t := \mathbb{E}(p_t - p_t^0). \quad (5)$$

The mean shift measures the amount (and direction) of deviation of the mean of the distribution under the considered path $(p_i)_{i=0}^T$, compared to mean $p_t^0$ at time $t$ if the distribution was not affected by the predictions.

**Analyzing the impact of PRM** We study the bias and shift of the PRM path $(\theta_t^*)_{t=0}^{T-1}$, which we denote by $\texttt{bias}_t^*$ and $\texttt{shift}_t^*$, respectively. Additionally, we compare the PRM path to a *naive path*, $\theta_t^n$, which ignores the performativity when making predictions. Formally, $\theta_t^n$ is defined as the mean of the previously observable distribution

$$\theta_t^n := p_{t-1}^{\text{test}}. \tag{6}$$

This corresponds to the usual approach to prediction in which one minimizes the loss with respect to the currently observable distribution (akin to the usual ERM principle). In the rapid case, where no distribution is observed in the first period, we define $\theta_0^n = 0$. We will denote the bias and shift of the naive path by $\texttt{bias}_t^n$ and $\texttt{shift}_t^n$, respectively.

**Interpreting bias and shift** Sections 4 and 5 derive explicit formulas for our impact measures under the PRM and naive paths, allowing us to reason about the quantitative behaviour of these metrics. In general, high bias can be interpreted as an undesirable property, even from a solely statistical standpoint (Young et al., 2005). However, as we will see, the PRM path is biased due to the trade-off with distribution uncertainty (Lemma 3.2). In contrast, mean shift interpretation is usually context-dependent. Section 6 provides two example scenarios to illustrate these points.

# 4. One-Period Model

This section analyzes the case of $T = 1$. First, we discuss the full information case where $p_0$, $\alpha$, $\lambda$, and $\pi$ are known to the model provider. This allows us to separate the effects of PRM from the hardness of designing of algorithms that achieve PRM (due to exploration/finite-sample effects). Next, we assume that the initial mean $p_0$ is unknown and study how this uncertainty affects our previous results. Finally, in an episodic RL setting, we study via simulations the case where no information about the parameters is available.

Notice that the slow deployment case for $T = 1$, which is the main focus of this section, corresponds to the standard setting of (Perdomo et al., 2020).

## 4.1. Perfect Information

### 4.1.1. SLOW DEPLOYMENT

**Proposition 4.1** (Proof in Appendix B.2)**.** *The solution to the problem (1) in the $T = 1$ slow deployment case is*

$$\theta_0^* = \begin{cases} \text{clip}\left(\frac{(1-|\alpha|)s_1}{1-2\alpha}, -\frac{1}{2}, \frac{1}{2}\right), & 1 - 2\alpha > 0, \\ \text{sign}(s_1)/2, & 1 - 2\alpha \leq 0. \end{cases}$$

*Whenever $|\theta_0^*| \neq 1/2$, we get*

$$p_1^* = \frac{(1-\alpha)(1-|\alpha|)}{1-2\alpha}s_1 = \frac{1-\alpha}{1-2\alpha}(\beta p_0 + (1-|\alpha|-\beta)\pi).$$

We visualize this solution in Figure 1. For the rest of the subsection we assume that $\theta_0^* \neq 1/2$.

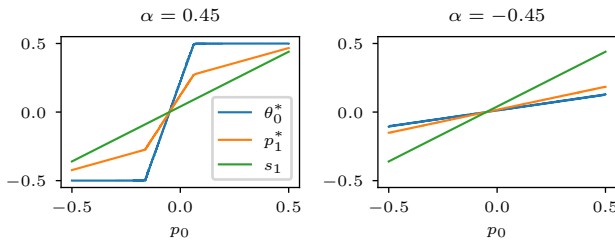

*Figure 1.* The dependence of $\theta_0^*$ (blue), $p_1^*$ (orange), and $s_1$ (green) on $p_0$ for $\lambda = 0.8$ and $\pi = 0.2$ in slow $T = 1$ case. Columns correspond to the different $\alpha$.

**Loss** We get that

$$\texttt{loss}_0^* = \begin{cases} \frac{1}{4} - \frac{(1-|\alpha|)^2 s_1^2}{1-2\alpha}, & |s_1| < \frac{1/2-\alpha}{1-|\alpha|}, \\ \frac{1-\alpha}{2} - (1-|\alpha|)|s_1|, & |s_1| \geq \frac{1/2-\alpha}{1-|\alpha|}. \end{cases}$$

**Bias** We have that $\texttt{bias}_0^* = \frac{\alpha(1-|\alpha|)s_1}{1-2\alpha}$. Thus, the PRM path is generally biased. This bias does not arise from the usual bias-variance trade-off in statistics. Instead, the performativity incentivizes the model provider to reduce the uncertainty in the distribution. To see this, notice that the unbiased predictor, which minimizes the error term in MSE (2), exists: $\theta_0^u = \frac{1-|\alpha|}{1-\alpha}s_1$. For positive feedback, the prediction is biased towards extreme values (i.e., $-1/2$ or $1/2$). For negative, the prediction is biased towards $0$. The absolute value of bias increases in $|\alpha|$ if $\alpha > -\frac{\sqrt{3}-1}{2}$.

**Shift** Once $\theta_0^*$ is deployed, it induces shift $\texttt{shift}_1^* = \frac{\alpha-|\alpha|+\alpha|\alpha|}{1-2\alpha}s_1$. The direction of the shift depends on $\text{sign}(\alpha)$ in the same way as the bias. The effect increases with $|\alpha|$. While a no-shift prediction exists, $\theta_0 = \text{sign}(\alpha)s_1$, it differs from the unbiased prediction in the negative feedback case. This shows that unbiasedness and the absence of shift can not be achieved simultaneously under negative feedback.

**Discussion** We can see that, in general, the PRM prediction is biased (even though the model provider has perfect information about the distribution), and its impact on the mean of the distribution is not zero. In the positive feedback case, the model provider benefits from shifting the mean to extreme values. Even though this strategy increases the error term, it decreases the uncertainty. In the negative feedback case, the performative response to the unbiased prediction shifts the mean closer to $0$. So, the provider employs a biased prediction to reduce this drop in the uncertainty.

**Comparison with Naive Path** Now, we consider the naive path, where, for the clarity of exposition, we assume that the system was initially at equilibrium, i.e.,

$p_0 = \pi = s_1$. We get

$$p_1^n = (1 + \alpha - |\alpha|)s_1,$$
$$\texttt{loss}_0^n = 1/4 - (1 + 2\alpha - 2|\alpha|)s_1^2,$$
$$\texttt{shift}_1^n = -\texttt{bias}_1^n = (\alpha - |\alpha|)s_1.$$

If $\alpha > 0$, the bias and shift of the naive path is zero, which might be more desirable compared to the PRM path. However, the naive loss is worse than the PRM loss by $\frac{\alpha^2}{1-2\alpha}s_1^2$. At the same time, if $\alpha \leq 0$, the bias and shift of the naive path are higher in absolute values than the bias and shift of the PRM path, i.e. RM increases our measures in the negative feedback case compared to PRM. Moreover, in the negative case, the loss penalty increases to $\frac{9\alpha^2}{1-2\alpha}s_1^2$.

### 4.1.2. RAPID DEPLOYMENT

**One-Period Case** Equation (1) reduces to

$$\min_{\theta_0 \in [-1/2, 1/2]} \theta_0^2 - 2\theta_0 p_0,$$

which results in $\theta_0^* = p_0$. (We visualize this solution in Figure 6, top row, in Appendix.) Additionally, we get

$$p_1^* = (\alpha + \beta)p_0 + (1 - |\alpha| - \beta)\pi,$$
$$\texttt{bias}_0^* = 0,$$
$$\texttt{shift}_1^* = (\alpha - |\alpha|\lambda)p_0 - |\alpha|(1 - \lambda)\pi.$$

If $\alpha > 0$, the PRM prediction shifts the mean closer to $p_0$ relative to $\pi$. If $\alpha < 0$, the effect is hard to interpret. We only consider the case of $\pi = p_0$. In this case, $\text{sign}(p_0) \neq \text{sign}(p_1^* - s_1)$. The mean is shifted away from $p_0$ in the direction of $0$. Also note that the absolute value of the shift increases in $|\alpha|$ under both negative and positive feedback.

**Comparison with Two-Period Case** To see whether the prediction remains unbiased once the distribution changes, we compare the one- and two-period models. For $T = 2$, we get the following two-period problem:

$$\min_{\theta_0, \theta_1, p_1 \in [-1/2, 1/2]} \sum_{t=0}^{1} \gamma^t(\theta_t^2 - 2\theta_t p_t)$$

such that $p_1 = \alpha\theta_0 + \beta p_0 + (1 - |\alpha|)(1 - \lambda)\pi$.

**Proposition 4.2** (Proof in Appendix B.3). *The solution to the problem (1) in the $T = 2$ rapid deployment case is*

$$\theta_0^* = \text{clip}\left(\frac{(1 + \gamma\alpha\beta)p_0 + \gamma\alpha(1 - |\alpha| - \beta)\pi}{1 - \gamma\alpha^2}, -\frac{1}{2}, \frac{1}{2}\right),$$
$$\theta_1^* = p_1^*.$$

*Whenever $|\theta_0^*| \neq 1/2$, we get*

$$p_1^* = \frac{(\alpha + \beta)p_0 + (1 - |\alpha| - \beta)\pi}{1 - \gamma\alpha^2}.$$

Figure 6, middle row, in Appendix visualizes $\theta_0^*$. If $|\theta_0^*| < 1/2$, we get

$$\texttt{bias}_0^* = \frac{\gamma\alpha(\alpha + \beta)p_0 + \gamma\alpha(1 - |\alpha| - \beta)\pi}{1 - \gamma\alpha^2},$$
$$\texttt{shift}_1^* = \frac{(\alpha - |\alpha|\lambda + \gamma\alpha^2\lambda)p_0 - (|\alpha| - \gamma\alpha^2)(1 - \lambda)\pi}{1 - \gamma\alpha^2},$$
$$\texttt{bias}_1^* = 0.$$

Compared to the case of $T = 1$, the mean shifts to more extreme values due to the denominator, and the first-period bias becomes non-zero. However, the final prediction remains unbiased. The long-term loss incentivizes the model provider to actively manipulate the mean, even if the short-term loss suffers from such manipulation.

**Summary** Similarly to the slow case, the bias and shift of the PRM path are generally not zero and increase in $|\alpha|$. In contrast to the slow case, only the long-term effects incentivize uncertainty optimization in the rapid model.

### 4.2. Imperfect Information

This section analyzes how uncertainty affects our full information results in the slow deployment case.

### 4.2.1. UNKNOWN MEAN

First, we analyze the case where $\alpha$ and $\lambda$ are known to the model provider but $p_0$ is unknown. Thus, the model provider needs to simultaneously learn $p_0$ and adjust for performativity. For simplicity, we focus on the equilibrium case where $p_0 = \pi$. To learn $p_0$, the model provider observes $m$ i.i.d. samples $S_0 = \{p_{0,i}\}_{i=1}^m \sim D_0^m$ and uses an estimator $\theta_0 : \mathbb{R}^m \to [-1/2, 1/2]$ to get an estimate $\theta_0(S_0)$.

**Estimators** To study the extent to which the results of the previous section transfer, we introduce the analogues of the optimal and naive predictions. For the naive case, we use the empirical mean $\hat{\theta}_0^n := \frac{1}{m}\sum_{i=1}^m p_{0,i} =: \bar{p}_0$. For the optimal case, we use the result from Proposition 4.1, in which we replace $s_1$ with $\bar{p}_0$, which results in estimator $\hat{\theta}_0^*$.

**Bias and Shift** Figure 2 depicts the bias and shift of $\hat{\theta}_0^*$ with one standard deviation confidence intervals. For $\alpha > 0$, the confidence intervals shrink very fast with $m$ for big values of $p_0$ due to the shrinking introduced by clip function.

**Loss** Now, we analyze the loss of $\hat{\theta}_0^*$.

**Theorem 4.3.** *The expected loss of $\hat{\theta}_0^*$ for $\alpha \leq 0$ is*

$$\mathbb{E}_{z \sim D_1^*}((\hat{\theta}_0^* - z)^2) = \frac{(1 - |\alpha|)^2}{1 - 2\alpha}\left(\frac{1 - 4(m+1)p_0}{4m}\right) + \frac{1}{4}.$$

*For all values of $\alpha$, the expected loss converges to the optimal expected loss:* $\lim_{m \to \infty} \mathbb{E}((\hat{\theta}_0^* - z)^2) = \mathbb{E}((\theta_0^* - z)^2)$.

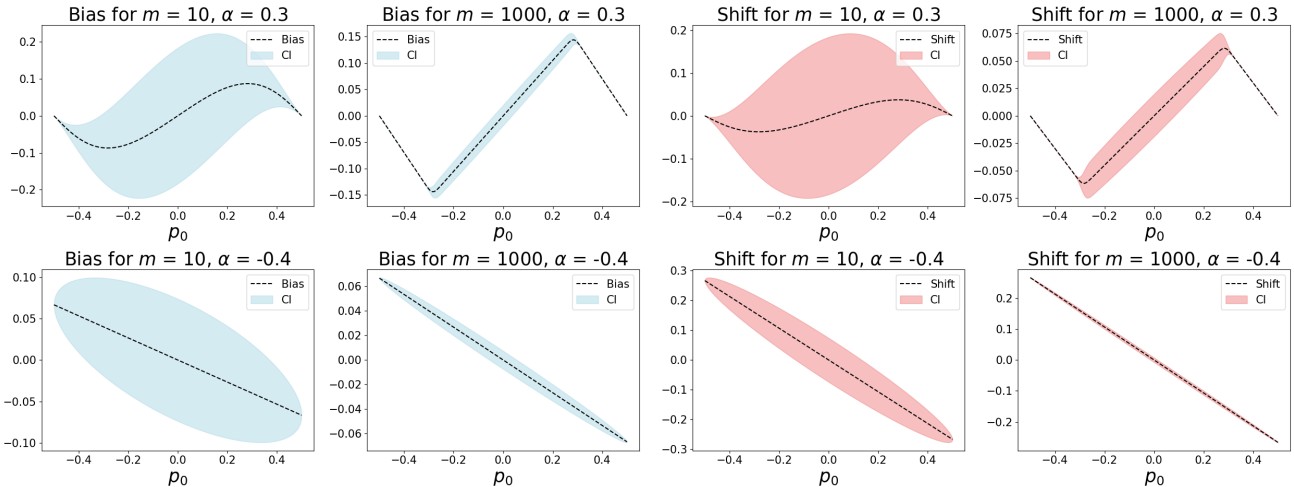

*Figure 2.* The dependence of $\mathtt{bias}(\hat{\theta}_0^*)$ (left) and $\mathtt{shift}(\hat{\theta}_0^*)$ (right) and corresponding variances on $p_0$. The upper row corresponds to $\alpha = 0.3$, the lower row corresponds to $\alpha = -0.4$. Columns correspond to the different $m$.

We discuss the loss for all values of $\alpha$ in Appendix B. To visualize the results of Theorem 4.3, we plot the difference in expected losses between $\hat{\theta}_0^*$ and $\hat{\theta}_0^n$ in Figure 3. Due to random sampling, we observe a region where $\hat{\theta}_0^n$ outperforms $\hat{\theta}_0^*$. However, for larger values of $m$, this region diminishes.

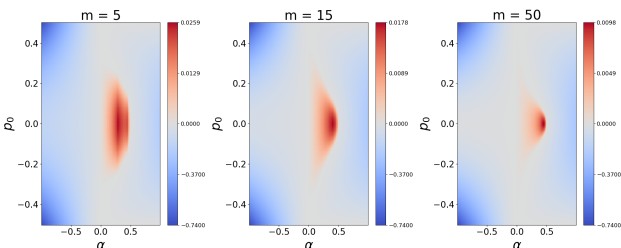

*Figure 3.* The dependence of the differences in expected losses, $\mathbb{E}(\mathtt{loss}(\hat{\theta}_0^*) - \mathtt{loss}(\hat{\theta}_0^n))$, on $p_0$ and $\alpha$, for different $m$.

### 4.2.2. RL SIMULATIONS

In this section, we check whether our results in the perfect information case transfer to the general performative prediction problem with information restrictions. In this setting, we consider episodic exploration and additionally assume that $\lambda = 0$ is known to the provider. In this case, the samples from the second period after deployment allow the model provider to estimate the performativity parameters.

We implement Algorithm 1 of Liu et al. (2022) with hyperparameter $\beta = 2^{-8}$ to find the optimal predictions. We visualize the prediction path of the algorithm in Figure 4 (left). After some exploration episodes, the predictions of the model provider and the means of the distribution quickly converge to the theoretically predicted values, which validates our results in the perfect information case.

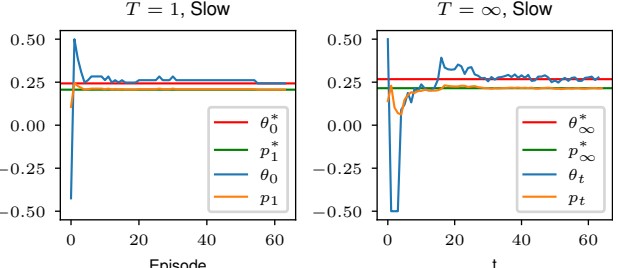

*Figure 4.* The predictions, $\theta_t$, (blue) the means, $p_t$, (orange) and their theoretical equilibrium values (red and green, respectively) in RL setting over episodes (left) or time (right) for $\pi = 0.2$, $\alpha = 0.15$, $\gamma = 0.9$, and $m = 100$, where $m$ is the number of samples observed from test distribution at each step. The left and right plots correspond to the $T = 1$ slow episodic setting (with $\lambda = 0$) and $T = \infty$ slow setting (with $\lambda = 0.3$), respectively.

## 5. Infinite Horizon Model

Now, we study the long-term effects of performativity by analyzing our model for $T = \infty$. We first theoretically study the perfect information case and then use simulations to analyze the case of unknown problem parameters.

### 5.1. Perfect Information

#### 5.1.1. SLOW DEPLOYMENT

**Theorem 5.1** (Proof in Appendix B.5)**.** *Assume that the PRM path does not take extreme values $\forall t \, |\theta_t^*| \neq 1/2$ and $1 - 2\alpha \geq \sqrt{\gamma}\beta$. Then, the solution to the problem (1) in the $T = \infty$ slow deployment case satisfies*

$$\frac{\theta_t^* - \theta_\infty^*}{p_0 - p_\infty^*} = \frac{2(1 - |\alpha|)\lambda}{1 - 2\alpha + \xi}\omega^t, \quad \frac{p_t^* - p_\infty^*}{p_0 - p_\infty^*} = \omega^t, \text{ where}$$

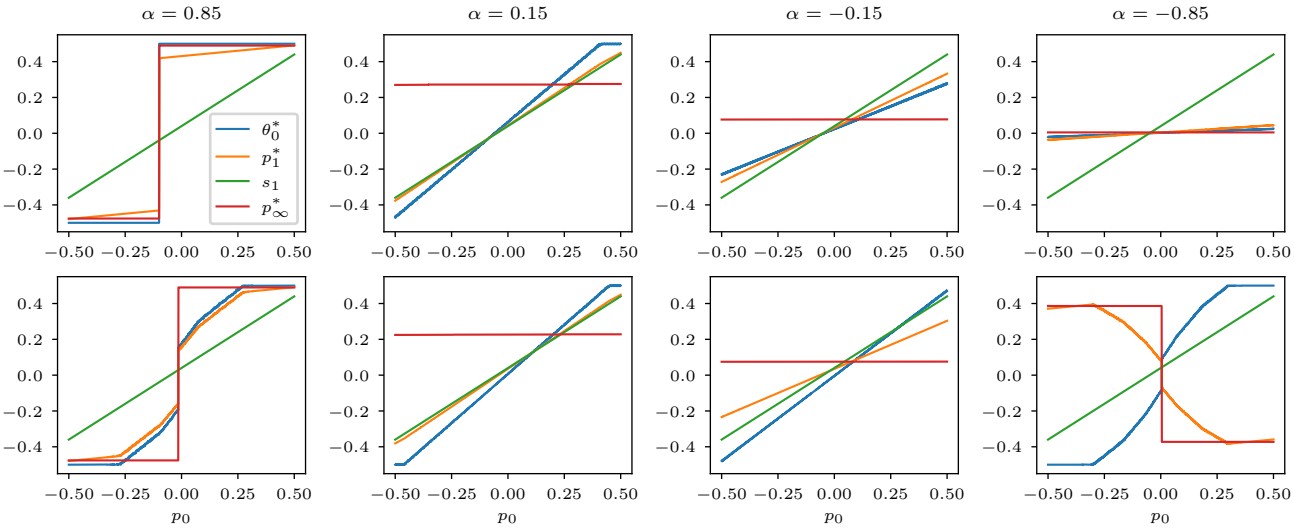

*Figure 5.* The plots depict the dependence of $\theta_0^*$ (blue), $p_1^*$ (orange), $s_1$ (green), and $p_\infty^*$ (red) on $p_0$ for $\lambda = 0.8$, $\pi = 0.2$, and $\gamma = 0.5$ in $T = \infty$ case. Columns correspond to the different $\alpha$, the top and bottom rows correspond to the slow and rapid cases, respectively.

$$\theta_\infty^* := \frac{(1-\gamma\beta)(1-|\alpha|-\beta)\pi}{1-2\alpha-\beta+\alpha\beta-\gamma\beta(1-\alpha-\beta)},$$

$$p_\infty^* := \frac{(1-\alpha-\gamma\beta)(1-|\alpha|-\beta)\pi}{1-2\alpha-\beta+\alpha\beta-\gamma\beta(1-\alpha-\beta)},$$

$$\omega := \beta + \frac{2\alpha\beta}{1-2\alpha+\xi}, \xi := \sqrt{1 - \frac{4\alpha(1-\alpha)}{1-\gamma\beta^2}}.$$

Notice that the restriction $\forall t \, |\theta_t^*| \neq 1/2$ could hold only if $\omega \leq 1$. There is an upper bound on $\alpha$ beyond which the model provider is incentivized to choose the extreme values of $\theta_t$. So, if this bound does not hold, after some time, the model provider always benefits from setting $|\theta_t^*| = 1/2$, even though this prediction is necessarily biased. Additionally, if $\omega < 1$, the solution converges $\theta_t^* \to \theta_\infty^*, s_t^* \to s_\infty^*, p_t^* \to p_\infty^*$ in the limit $t \to \infty$, allowing us to study the long-term effects of PRM.

We visualize the solution for all cases in Figure 5 (top row). The restriction $\forall t \, |\theta_t^*| \neq 1/2$ does not cover the cases of big positive values of $\alpha$. In such scenarios, the PRM prediction depends on $p_0$ discontinuously because the model provider has a strong incentive to shift the mean to extreme values.

For the rest of this section, we assume that $\pi > 0$.

**Long-Term Bias**  The long-term bias follows

$$\theta_\infty^* - p_\infty^* = \frac{\alpha(1-|\alpha|-\beta)\pi}{1-2\alpha-\beta+\alpha\beta-\gamma\beta(1-\alpha-\beta)}.$$

Even in the limit $t \to \infty$, the PRM solution has a non-vanishing bias. If $\alpha > 0$ and $\alpha$ is small, the long-term bias is positive. Even though the bias increases the error term in Equation (2), the model provider benefits in terms of

uncertainty because the biased prediction shifts the mean to more extreme values. On the other hand, if $\alpha < 0$ and $|\alpha|$ is small, the bias is negative. In the negative feedback case, the negative bias again shifts the mean to more extreme values than the unbiased prediction, reducing uncertainty.

**Long-Term Shift**  The long-term shift of $\theta_t^*$ is non-zero:

$$p_\infty^* - \pi = \frac{(\alpha - |\alpha| + \alpha|\alpha| + \gamma\beta(|\alpha| - \alpha))\pi}{1-2\alpha-\beta+\alpha\beta-\gamma\beta(1-\alpha-\beta)}.$$

**Comparison with Naive Path**  We have that $\theta_\infty^n = p_\infty^n = \frac{1-|\alpha|-\beta}{1-\alpha-\beta}\pi$. The bias of the naive path tends to zero as $t \to \infty$. The long-term shift is also zero if $\alpha > 0$. If $\alpha < 0$,

$$\frac{p_\infty^* - \pi}{p_\infty^n - \pi} = \frac{1 - \gamma\beta + \frac{|\alpha|}{2}}{1 - \gamma\beta + \frac{|\alpha|}{1+|\alpha|/(1-\beta)}} < 1.$$

The long-term shift of the naive path is bigger than that of the PRM path in the negative feedback case.

Similarly to $T = 1$, the naive path has a smaller bias and shift than the PRM path in the positive feedback case, while the PRM path has a smaller shift in the negative feedback case. However, the long-term bias of the naive path is $0$, even in the negative feedback case.

### 5.1.2. RAPID DEPLOYMENT

**Theorem 5.2** (Proof in Appendix B.6). *Assume that the PRM path does not take extreme values $\forall t \, |\theta_t^*| \neq 1/2$. Then, the solution to the problem (1) in $T = \infty$ rapid case satisfies*

$$\theta_t^* = \frac{2}{1+\chi}(p_0 - p_\infty^*)\kappa^t + \theta_\infty^*, \; p_t^* = (p_0 - p_\infty^*)\kappa^t + p_\infty^*,$$

*where* $\kappa := \beta + \frac{2\alpha}{1+\chi}$, $\chi := \sqrt{1 - \frac{4\gamma\alpha(\alpha+\beta)}{1-\gamma\beta^2}}$ *and*

$$\theta_\infty^* := \frac{(1-\gamma\beta)(1-|\alpha|-\beta)\pi}{1-\alpha-\beta-\gamma(\alpha+\beta-\beta(2\alpha+\beta))},$$

$$p_\infty^* := \frac{(1-\gamma(\alpha+\beta))(1-|\alpha|-\beta)\pi}{1-\alpha-\beta-\gamma(\alpha+\beta-\beta(2\alpha+\beta))}.$$

Similarly to the slow case, the restriction $\forall t \, |\theta_t^*| \neq 1/2$ could hold only if $|\kappa| \leq 1$. If $|\kappa| < 1$, $\theta_\infty^*$ and $p_\infty^*$ represent the long-term values of $\theta_t^*$ and $p_t^*$, respectively.

Figure 5 (bottom row) visualizes the solution for all cases. Again, the assumption $\forall t |\theta_t^*| \neq 1/2$ does not cover large $|\alpha|$. If $|\alpha|$ is large, the PRM prediction depends discontinuously on $p_0$. If $\alpha > 0$, the mean, depending on $p_0$, converges to one of two equilibrium values. If $\alpha < 0$, the mean oscillates between two values that correspond to extreme predictions.

For the rest of this section, we assume that $\pi > 0$.

**Long-Term Bias** We get

$$\theta_\infty^* - p_\infty^* = \frac{\gamma\alpha(1-|\alpha|-\beta)\pi}{1-\alpha-\beta-\gamma(\alpha+\beta-\beta(2\alpha+\beta))}.$$

The bias is again not zero and behaves similarly to the slow case for small $|\alpha|$.

**Long-Term Shift** We get a non-zero long-term shift:

$$p_\infty^* - \pi = \frac{(\alpha-|\alpha|+\gamma(\alpha|\alpha|+(|\alpha|-\alpha)\beta)\pi}{1-\alpha-\beta-\gamma(\alpha+\beta-\beta(2\alpha+\beta))}.$$

**Comparison with Naive Path** Notice that the mean in the naive path case satisfies $p_{t+1}^n = \alpha p_{t-1}^n + \beta p_t^n + (1-|\alpha|-\beta)\pi$. Since $\alpha + \beta < 1$, the mean converges to an equilibrium, which satisfies $\theta_\infty^n = p_\infty^n = \frac{1-|\alpha|-\beta}{1-\alpha-\beta}\pi$. Again, the long-term bias of the naive path is zero. The shift is zero if $\alpha > 0$. If $\alpha < 0$,

$$\frac{p_\infty^* - \pi}{p_\infty^n - \pi} = \frac{1+\gamma|\alpha|/2 - \gamma\beta}{1+\gamma|\alpha|/2 - \gamma\beta + \frac{\gamma|\alpha|(1-|\alpha|-\beta)}{2(1+|\alpha|-\beta)}} < 1.$$

Similarly to the slow case, the shift is smaller for $\theta_t^*$.

### 5.2. RL Simulations

Finally, we check whether our results in the perfect information case transfer to the general performative prediction problem with information restrictions. We consider a usual sequential RL problem. We implement a simple heuristic algorithm, which learns the performative response by deploying extreme predictions $\{-1/2, 1/2\}$ at random for the first 4 steps. Then the model provider learns the parameters of the performative response by likelihood maximization and deploys the optimal policy under the resulting

estimates. We visualize the prediction path of the algorithm in Figure 4 (right). After some exploration, the predictions and the means of the distribution quickly converge to the theoretically-predicted equilibrium values, which validates our theoretical analysis of the perfect information case.

## 6. Discussion and Future Work

Our results suggest that the performatively optimal (PRM) path is, in general, biased and introduces a non-zero mean shift. These effects are more expressed when the mean responds positively to model predictions or when it responds negatively, but the model is updated rapidly and the performativity is high. To understand the potential impact of such effects, we now provide two example scenarios and interpret our measures and technical results in a social context.

**Case study: drug efficacy estimation** Consider a scenario in which a company is trying to estimate the effectiveness of a drug they produce against a specific disease. We define our binary random variables as indicators that the drug cures a randomly sampled patient. To model the well-known placebo effect, under which beliefs about the effectiveness of a drug may further increase its positive impact, we assume a positive performative response ($\alpha > 0$). Consider the one-period positive feedback model in Section 4. Then, a positive/negative bias indicates an exaggerated/understated prediction of the average drug efficacy respectively, which may make it harder to find the most effective drug on the market. At the same time, a positive shift indicates a higher drug efficacy due to the placebo effect, which is, of course, desirable for combating the disease.

Our results in Section 4 with $p_0 = \pi$ suggest that whenever $p_0 > 0$ (i.e. the drug is effective to begin with), PRM would lead to a positive bias, i.e. exaggerated prediction on the drug's effectiveness; as well as positive shift and thus increased drug effectiveness due to performativity.

**Case study: traffic prediction** Consider a model provider seeking to predict which of the two roads, A or B, is less busy. We model this by defining the binary random variable as an indicator for the event that road A is less busy. Consider our infinite horizon negative feedback model. Positive or negative bias of PRM corresponds to the model provider redirecting more traffic to road A or B respectively. At the same time, positive or negative shift indicates an increase in the usage of road B or A respectively. The bias is probably an undesirable property of the prediction as it makes some drivers choose a sub-optimal road. At the same time, the shift might be benign or adverse, depending on the context.

In the slow deployment case, the mean usage of roads becomes more equalized (Figure 5, top-right part) compared to the case when no performativity is present, which is in-

tuitively desirable. In the rapid deployment case, if the strength of performativity is small, the usage becomes more equalized (Figure 5, bottom row, third plot). If the performativity is large, the usage oscillates between roads (Figure 5, bottom row, fourth plot), which may be undesirable.

**Limitations and future work**   In this work, we focus on mean estimation of binary variables only and work under the linear response model (3). This makes the analysis of the long-term dynamics driven by (1) tractable and allows for defining natural metrics of impact and interpreting them in context. Despite its simplicity, we hope that our model can be qualitatively useful in broader settings. First, the linear response naturally arises as a first-order Taylor approximation for any performative response. Thus, our results may (at least qualitatively) transfer to situations of weak performative response. Second, as noted in the discussion of Lemma 3.2, the error-uncertainty decomposition holds for a broad class of distributions. Thus, we can expect PRM to generally prefer distributions with smaller aleatoric uncertainty. For example, in the case of multinomial distribution, the model provider has an additional incentive to concentrate the probability mass on a small subset of outcomes.

Additionally, our results can easily be extended to the following more general group setting. Imagine that clients consist of several independent groups, and each group reacts to the predictions of the model in the same way as the whole distribution in our paper. Also, assume that the model provider additionally observes covariates that are predictive for group membership before making a prediction. This modification makes our problem much closer to the usual supervised learning tasks where the model provider needs to simultaneously learn a model for membership prediction and outcomes for each group. At the same time, our results in the perfect information setting can be directly transfered to this setup by independently applying the previous analysis to each group. The main limitation of such an extension is the assumption that groups evolve independently. This assumption could hold in the setting of drug efficacy prediction, but it will probably not hold in traffic prediction.

We hope that our work will encourage further analysis of the broader impact of PRM . In particular, it would be interesting to analyze more complex distributions (e.g., in a regression setting) and models of performative response.

## Acknowledgments

This research was partially funded from the Ministry of Education and Science of Bulgaria (support for INSAIT, part of the Bulgarian National Roadmap for Research Infrastructure). The authors thank Kristian Minchev for his helpful feedback and discussions on this work.

## Impact Statement

This paper presents work whose goal is to advance the field of Machine Learning. Our theoretical analysis contributes to better understanding of the impact of machine learning on society. Therefore, we expect that our results can serve a positive purpose in increasing the awareness about ML impact and encouraging further research on related topics. There are many potential societal consequences of our work, none which we feel must be specifically highlighted here.

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

# Supplementary Material

- Appendix A contains additional results.

- Appendix B contains proofs of all results.

- Appendix C contains details of RL experiments.

## A. Additional Results

This section presents additional results that were not included in the main text.

### A.1. Generalization of Impact Metrics

This section discusses possible generalizations of the impact metrics defined in Section 3.3

**Mean Shift Metric**    Regarding the mean shift metric, we identify two possible extensions, for parametric and non-parametric distributions, respectively.

Consider a setting where the distribution is parametrized by a finite number of parameters, $D_t = D(\mathbf{w}_t)$, where $\mathbf{w}_t = (w_t^1, \dots, w_t^k)$. This is, for example, the case for distributions defined via causal graphical models, as well as for many common distributions (e.g., exponential families). Then, one can define the parameter shift metric as $\mathtt{shift}_t = \mathbb{E}(\mathbf{w}_t - \mathbf{w}_t^0)$, where $\mathbf{w}_t^0$ are the parameters of the distribution at time $t$ if the distribution was not affected by the performativity.

In a non-parametric setting, we can instead define a divergence-based metric $\mathtt{shift}_t = \mathbb{E}(K(D_t, D_t^0))$, where $K$ is an arbitrary divergence function (e.g., KL-divergence). The function $K$ can be designed to capture an undesirable shift in the distribution, according to the target application.

These metrics can then be studied under different models for the performative response and counterfactual dynamics of the distribution in the absence of performativity, which can be chosen depending on the learning task and application under consideration.

**Bias Metric**    Regarding the bias metric, we provide two extensions suitable for cases where the distribution is divided into several groups, which is relevant, e.g., in fairness-sensitive applications.

First, in a setting with several groups and multi-labeled data, one could calculate a matrix of biases with one entry for each group and label defined as follows $\mathtt{bias}^{g,y} = \mathbb{E}(\mathbb{E}_{(X,Y,G)\sim D_t^{test}}(q_t^y(X) - [Y = y]|G = g))$, where $q_t(X) = (q_t^1(X), \dots, q_t^{|\mathcal{Y}|}(X))$ is the vector of model's softmax probabilities at time $t$ and $G$ is the group. These biases could be interpreted as a measure of unfairness among groups.

Second, one can use established metrics from the literature of bias amplification see Zhao et al. (2017); Wang & Russakovsky (2021); Zhao et al. (2023); Tokas et al. (2024).

### A.2. Perfect Information

This section contain additional results in the perfect information setting.

#### A.2.1. ONE-PERIOD SLOW DEPLOYMENT, PERFECT INFORMATION

Here, we present additional results for Section 4.1.1.

**Comparison with Naive Path in Symmetric Case** In the symmetric case, we assume that the equilibrium probability is symmetric, $\pi = 0$. Then,

$$\texttt{loss}_0^n = 1/4 + (1 - 2\alpha - 2\beta)p_0^2, \quad \texttt{loss}_0^* = 1/4 - \frac{\beta^2}{1 - 2\alpha}p_0^2,$$

$$\texttt{bias}_0^n = (1 - \alpha - \beta)p_0, \quad\quad \texttt{bias}_0^* = \frac{\alpha\beta}{1 - 2\alpha}p_0,$$

$$\texttt{shift}_1^n = (\alpha - |\alpha|\lambda)p_0, \quad\quad \texttt{shift}_1^* = \frac{(\alpha - |\alpha| + \alpha|\alpha|)\lambda}{1 - 2\alpha}p_0.$$

If $\alpha > 0$, we get

$$\frac{|\texttt{bias}_0^*|}{|\texttt{bias}_0^n|} = \frac{\alpha\lambda}{(1 - 2\alpha)(1 - \lambda)}.$$

If the performativity, $\alpha$, or the inertia, $\lambda$, is big then the naive prediction is preferable in terms of bias. Otherwise, the optimal prediction is preferable. (The same analysis holds for the shift of estimator.)

If $\alpha \leq 0$, we get

$$\frac{|\texttt{bias}_0^*|}{|\texttt{bias}_0^n|} = \frac{|\alpha|(1 - |\alpha|)\lambda}{(1 + 2|\alpha|)(1 - \lambda + |\alpha| + |\alpha|\lambda)} < 1.$$

Thus, the optimal prediction is preferable in terms of bias. (The same is true for the shift.)

Finally, if $1 - 2\alpha > 0$, the loss penalty equals to

$$\left(1 - 2\alpha - 2\beta + \frac{\beta^2}{1 - 2\alpha}\right)p_0^2.$$

To analyze it consider two cases: $\lambda = 0$ and $\lambda = 1$. If $\lambda = 0$, we get that the following penalty

$$\texttt{loss}_0^n - \texttt{loss}_0^* = (1 - 2\alpha)p_0^2.$$

This penalty is bigger than the penalty in the equilibrium case for small $\alpha$. If $\lambda = 1$, we get that $p_0 = s_1$. So, we get the same answer as in the equilibrium case.

### A.2.2. TWO-PERIOD SLOW DEPLOYMENT, PERFECT INFORMATION

Here, extend Section 4.1.1 by solving the two-period case and comparing with it.

**Proposition A.1** (Proof in Appendix B.7). *Assume that $1 - 2\alpha > \sqrt{\gamma}|\alpha|\beta$. Then, the solution to the problem (1) in $T = 2$ slow case satisfies*

$$\theta_0^* = \mathrm{clip}\left(\frac{(1 - |\alpha|)((1 - 2\alpha + \gamma\alpha\beta^2)s_1 + \gamma\alpha\beta(1 - \lambda)\pi)}{(1 - 2\alpha)^2 - \gamma\alpha^2\beta^2}, -\frac{1}{2}, \frac{1}{2}\right),$$

*if $2(1 - |\alpha|)|s_2^*| \leq 1 - 2\alpha$ (which always holds for $\alpha \leq 0$).*

We visualize whole solution on Figure 6, bottom row. Notice that on the left part of the picture we operate in regime $1 - 2\alpha < \sqrt{\gamma}|\alpha|\beta$. In this situation, the optimal prediction depends non-continuously on $p_0$ because of the incentive to push the mean to the extremes. Additionally notice that the left plot has a kink on its right side. This kink corresponds to the transition between the cases $2(1 - |\alpha|)|s_2^*| \leq 1 - 2\alpha$ and $2(1 - |\alpha|)|s_2^*| > 1 - 2\alpha$.

If $|\theta_0^*| < 1/2$ in the setting of Proposition A.1, we get

$$p_1^* = \frac{(1 - |\alpha|)((1 - 2\alpha)(1 - \alpha)s_1 + \gamma\alpha^2\beta(1 - \lambda)\pi)}{(1 - 2\alpha)^2 - \gamma\alpha^2\beta^2}.$$

For the rest of the subsubsection we assume $\theta_0^* < 1/2$.

**Bias of $\theta_0^*$** We get

$$\theta_0^* - p_1^* = \frac{\alpha(1 - |\alpha|)((1 - 2\alpha + \gamma\beta^2)s_1 + \gamma(1 - \alpha)\beta(1 - \lambda)\pi)}{(1 - 2\alpha)^2 - \gamma\alpha^2\beta^2}.$$

For equilibrium and symmetric $\pi$, the bias of prediction becomes more pronounced because

$$\frac{1 - 2\alpha + \gamma\beta^2}{(1 - 2\alpha)^2 - \gamma\lambda^2\beta^2} \geq \frac{1}{1 - 2\alpha}.$$

**Shift of $\theta_0^*$**  We get

$$p_1^* - s_1 = \frac{((1-2\alpha)(\alpha - |\alpha| + \alpha|\alpha|) + \gamma\alpha^2\beta^2)s_1 + \gamma\alpha^2(1-|\alpha|)\beta(1-\lambda)\pi}{(1-2\alpha)^2 - \gamma\alpha^2\beta^2}.$$

**Discussion**  We can see that generally the bias of the optimal prediction is exacerbated in the two-period model. It happens because the motivation of the model provider to skew the distribution becomes stronger due to longer horizon.

**Comparison with Naive Path**  In equilibrium case $\pi = p_0$, given that the impact and bias of the naive path is the same as for the one-period model and our results above, we get that the naive path is even more preferable to the optimal path if $\alpha > 0$ in terms of bias and shift. For the case of $\alpha < 0$, we get

$$-\frac{\texttt{bias}_0^*}{|\alpha|s_1} = -\frac{(1-|\alpha|)(1+2|\alpha| + \gamma\beta(1+|\alpha| - 2|\alpha|\lambda))}{(1+2|\alpha|)^2 - \gamma|\alpha|^2\beta^2},$$

$$\frac{\texttt{shift}_1^*}{|\alpha|s_1} = \frac{-(1+2|\alpha|)(2+|\alpha|) + \gamma|\alpha|(1-|\alpha|)\beta}{(1+2|\alpha|)^2 - \gamma|\alpha|^2\beta^2},$$

$$\frac{\texttt{shift}_1^n}{|\alpha|s_1} = -\frac{\texttt{bias}_0^n}{|\alpha|s_1} = -2.$$

By direct calculation, the bias and shift of the naive path is always higher than those of the optimal path.

In the symmetric case $\pi = 0$, if $\alpha > 0$, we get

$$\frac{|\texttt{bias}_0^*|}{|\texttt{bias}_0^n|} = \frac{\alpha\lambda(1 + \gamma\beta^2/(1-2\alpha))}{((1-2\alpha) - \gamma\alpha^2\beta^2/(1-2\alpha))(1-\lambda)}$$

Notice that ratio $\beta^2/(1-2\alpha) = \lambda^2(1 + \alpha^2/(1-2\alpha))$ is increasing in $\alpha$. Thus, the ratio of biases is increasing in $\alpha$ and $\lambda$. So, similarly, to the one-period case, the optimal path is preferable to the naive path in terms of bias if $\alpha$ and $\lambda$ are small enough. (Same analysis holds for the shift.)

If $\alpha < 0$, we get

$$\frac{|\texttt{shift}_1^*|}{|\texttt{shift}_1^n|} = \frac{|\alpha|\lambda(1+2|\alpha| + \gamma\beta^2)}{((1+2|\alpha|)^2 - \gamma\alpha^2\beta^2)(1 + |\alpha| - \lambda + |\alpha|\lambda)}.$$

This ratio is increasing in $\lambda$ and $\gamma$. Hence,

$$\frac{|\texttt{shift}_1^*|}{|\texttt{shift}_1^n|} \le \frac{2 + \alpha^2}{2 + 8|\alpha| + 3\alpha^2} \le 1.$$

So, the bias of the optimal path is smaller than the bias of the naive path. Similarly, the shift of the optimal path is smaller than the impact of the naive path.

**Discussion**  Similarly to the one-period case, the naive path might be preferable in terms of bias and impact to the optimal path for $\alpha \ge 0$. However, for $\alpha < 0$, the optimal path is superior to the naive path in terms of bias and shift.

### A.2.3. INFINITE HORIZON SLOW DEPLOYMENT, PERFECT INFORMATION

This section contains additional results for Section 5.1.

**Bias of $\theta_0^*$**  We get

$$\texttt{bias}_0^* = \frac{(1-\xi)(1-|\alpha|)}{1 - 2\alpha + \xi}s_1.$$

Notice that, if $1 - 2\alpha \ge \sqrt{\gamma}\beta$, then this bias is bigger than in the two-period case. Thus, as previously, the longer time horizon incentivizes the model provider to give more biased predictions.

**Shift of $\theta_0^*$**    We get

$$\texttt{shift}_1^* = \frac{2\alpha - |\alpha| - |\alpha|\xi}{1 - 2\alpha + \xi} s_1.$$

Similarly to the bias of $\theta_0^*$, the impact of $\theta_0^*$ increases compared to the two-period case if $\alpha > 0$. However, if $\alpha < 0$, the impact becomes smaller than the two-period impact.

**Bias and Shift of $\theta_0^n$**    The bias and impact of the naive path in the symmetric case follows

$$\texttt{bias}_0^n = (1 - \alpha - \beta)p_0,$$
$$\texttt{shift}_1^n = (\alpha - |\alpha|\lambda)p_0.$$

The bias of the naive path is smaller if

$$2\alpha + \beta(1 + \gamma(\alpha + \beta)(1 - \alpha - \beta)) \geq 1,$$

which happens only if $\alpha > 0$ and $\alpha$ is sufficiently big. (The same inequality holds for the shift.)

A.2.4. INFINITE HORIZON RAPID DEPLOYMENT, PERFECT INFORMATION

This section contains additional results for Section 5.1.2.

**Bias of $\theta_0^*$**    We get

$$\theta_0^* - p_0 = \frac{1 - \chi}{1 + \chi}p_0.$$

Assuming that $p_0 > 0$, we get the following classification of the model provider actions. In the case of $\alpha > 0$, we get that $\theta_0^* > p_0$. If $\alpha < 0$ and $\alpha + \beta > 0$, $\theta_0^* < p_0$. Finally, if $\alpha + \beta < 0$, $\theta_0^* > p_0$ again.

**Shift of $\theta_0^*$**    We get

$$p_1^* - s_1 = \kappa p_0 - \lambda p_0 = \left(-|\alpha|\lambda + \frac{2\alpha}{1 + \chi}\right)p_0.$$

Since $\kappa$ increases in $\alpha$ and $\kappa|_{\alpha=0} = \lambda$, the shift increases in $|\alpha|$.

A.2.5. ADDITIONAL VISUALIZATIONS

We visualize the solutions for $T = 1$ rapid case, $T = 2$ rapid case, and $T = 2$ slow case in Figure 6. As we can see, if $\alpha > 0$, the prediction and the resulting next-period mean shift to more extreme values. Otherwise, the prediction and mean shift to 0 (the effect is more pronounced for the mean).

**A.3. Imperfect Information, $T = 1$ Slow Deployment**

Here, we present additional results for Section 4.2.

A.3.1. BIAS AND MEAN SHIFT THEORETICAL RESULTS

The bias for the naive estimator $\hat{\theta}_0^n$ is given by

$$\texttt{bias}_0^n = p_0(|\alpha| - \alpha).$$

For the performative estimator $\hat{\theta}_0^*$, the bias is

$$\texttt{bias}_0^* = (1 - \alpha)\mathbb{E}[\hat{\theta}_0^*] - (1 - |\alpha|)p_0.$$

For the naive estimator $\hat{\theta}_0^n$ the mean shift is

$$\texttt{shift}_1^n = p_0(\alpha - |\alpha|) = -\texttt{bias}_0^n,$$

and for the performative estimator $\hat{\theta}_0^*$, we have

$$\texttt{shift}_1^* = \alpha\,\mathbb{E}[\hat{\theta}_0^*] - |\alpha|p_0.$$

A.3.2. GENERAL VERSION OF THEOREM 4.3

Here, we present a result that generalizes Theorem 4.3, offering theoretical insights for all possible values of $\alpha \in (-1, 1)$.

**Theorem A.2.** *For the naive estimator $\hat{\theta}_0^n$ the expected loss is*

$$\mathbb{E}_{z \sim D_1^{test}}[(\hat{\theta}_0^n - z)^2] = p_0^2(2|\alpha| - 2\alpha - 1) + (2\alpha - 1)\frac{4p_0^2 - 1}{4m} + \frac{1}{4},$$

*and for the performative estimator $\hat{\theta}_0^*$, we have*

$\mathbb{E}[(\hat{\theta}_0^* - z)^2] =$
$$\begin{cases} \frac{(1-|\alpha|)^2}{1-2\alpha}\left(\frac{\frac{1}{4} - p_0^2}{m} - p_0^2\right) + \frac{1}{4} & \alpha \in (-1, 0] \\ p_0(1 - |\alpha|)\left(2F_{m, p_0 + \frac{1}{2}}\left(\frac{m}{2}\right) - 1\right) + \frac{1-\alpha}{2} & \alpha \in [0.5, 1) \\ \sum_{x \in I}((1 - 2\alpha)g(x)^2 - 2(1 - |\alpha|)p_0g(x))p(x) + (p_0(1 - |\alpha|) - \frac{1-2\alpha}{4})F_{m, p_0 + \frac{1}{2}}\left(\frac{2-3\alpha}{2-2\alpha}m\right) \\ + (p_0(1 - |\alpha|) + \frac{1-2\alpha}{4})F_{m, p_0 + \frac{1}{2}}\left(\frac{\alpha m}{2-2\alpha}\right) - p_0(1 - |\alpha|) + \frac{1-\alpha}{2}, & \alpha \in (0, 0.5), \end{cases}$$

*where $I$ is the set of integers in $\left(\frac{\alpha m}{2-2\alpha}, \frac{(2-3\alpha)m}{2-2\alpha}\right]$, $g(x) := \left(\frac{1-\alpha}{1-2\alpha}\right)\left(\frac{x}{m} - \frac{1}{2}\right)$, $F_{m, p_0 + \frac{1}{2}}(x) := \sum_{k=0}^{\lfloor x \rfloor} p(x)$, and*

$$p(x) := \binom{m}{x}\left(\frac{1}{2} + p_0\right)^x \left(\frac{1}{2} - p_0\right)^{m-x}$$

*Asymptotically, we have that as $m \to \infty$*
$$\mathbb{E}[(\hat{\theta}_0^* - z)^2] \to \texttt{loss}_0^*$$

*i.e. as $m$ goes to infinity, $\hat{\theta}_0^*$ approaches the optimal estimator for the risk minimisation problem.*

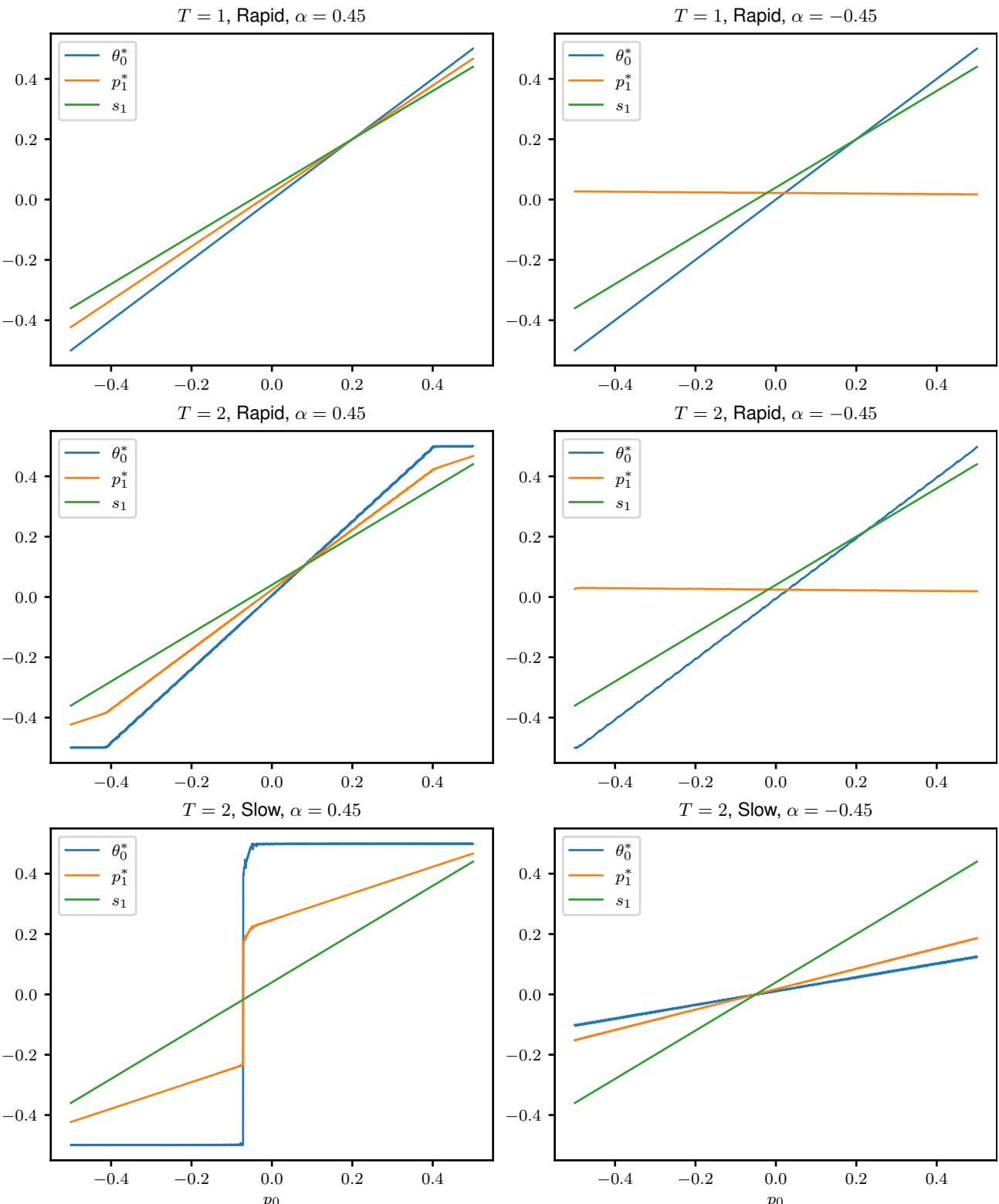

*Figure 6.* The plots depict the dependence of $\theta_0^*$ (blue), $p_1^*$ (orange), and $s_1$ (green) on $p_0$ for $\lambda = 0.8$, $\pi = 0.2$, and $\gamma = 0.5$. Columns correspond to the different values of $\alpha$; the top row corresponds to the $T = 1$ rapid case; the middle row corresponds to $T = 2$ rapid case; the bottom row corresponds to the $T = 2$ slow case.

# B. Proofs

## B.1. Proof of Lemma 3.2

*Proof.* Using conditional expectation we have

$$
\begin{aligned}
\mathbb{E}[(\theta_t - z)^2 \mid \theta_t, p_t^{test}] &= \mathbb{E}[\theta_t^2 - 2\theta_t z + z^2 \mid \theta_t, p_t^{test}] \\
&= \theta_t^2 - 2\theta_t \, \mathbb{E}[z \mid \theta_t, p_t^{test}] + \mathbb{E}[z^2 \mid \theta_t, p_t^{test}] \\
&= \theta_t^2 - 2\theta_t p_t^{test} + \frac{1}{4} \\
&= (\theta_t^2 - p_t^{test})^2 + \frac{1}{4} - (p_t^{test})^2
\end{aligned}
$$

$\square$

## B.2. Proof of Proposition 4.1

By direct calculation,

$$
\theta_0^2 - 2\theta_0(\alpha\theta_0 + (1 - |\alpha|)s_1) = (1 - 2\alpha)\theta_0^2 - 2(1 - |\alpha|)s_1\theta_0.
$$

If the parabola above opens downwards, $1 - 2\alpha \leq 0$, it achieves minimum at the extreme point of the domain. By analyzing both extreme points, we get $\theta_0^* = \text{sign}(s_1)/2$.

If the parabola opens upwards, $1 - 2\alpha > 0$, it achieves the minimum at the point in our domain closest to the vertex point. Thus, $\theta_0^* = \text{clip}\left(\frac{(1-|\alpha|)s_1}{1-2\alpha}, -\frac{1}{2}, \frac{1}{2}\right)$.

## B.3. Proof of Proposition 4.2

Going backward, we get

$$
\theta_1^* = p_1,
$$

which results in the following problem

$$
\min_{\theta_0, \theta_1, p_1} \theta_0^2 - 2\theta_0 p_0 - \gamma p_1^2 \text{ s.t. } p_1 = \alpha\theta_0 + (1 - |\alpha|)(\lambda p_0 + (1 - \lambda)\pi), \theta_t \in [-1/2, 1/2].
$$

Similarly to Appendix B.2, we get

$$
\theta_0^* = \text{clip}\left(\frac{(1 + \gamma\alpha\beta)p_0 + \gamma\alpha(1 - |\alpha|)(1 - \lambda)\pi}{1 - \gamma\alpha^2}, -\frac{1}{2}, \frac{1}{2}\right).
$$

(Notice that $1 - \gamma\alpha^2 > 0$, which reduces the number of cases.)

## B.4. Proof of Theorems 4.3 and A.2

Before computing the expected loss, we first show the following result regarding the first two moments of the performative estimator.

**Lemma B.1** (Moments of the Performative Estimator)**.** *For the performative estimator $\hat{\theta}_0^*$, we have that the first two moments are given by*

$$
\mathbb{E}[\hat{\theta}_0^*] = \begin{cases}
\frac{(1-|\alpha|)p_0}{1-2\alpha} & \alpha \in (-1, 0] \\
\frac{1}{2} - F_{m, p_0 + \frac{1}{2}}\left(\frac{m}{2}\right) & \alpha \in [0.5, 1) \\
\sum_{x \in I}\left(\frac{1-\alpha}{1-2\alpha}\right)\left(\frac{x}{m} - \frac{1}{2}\right)p(x) \\
+\frac{1}{2} - \frac{1}{2}F_{m, p_0 + \frac{1}{2}}\left(\frac{2-3\alpha}{2-2\alpha}m\right) \\
-\frac{1}{2}F_{m, p_0 + \frac{1}{2}}\left(\frac{\alpha}{2-2\alpha}m\right) & \alpha \in (0, 0.5)
\end{cases}
$$

*and*

$$\mathbb{E}[(\hat{\theta}_0^*)^2] = \begin{cases} \left(\frac{1-|\alpha|}{1-2\alpha}\right)^2\left(\frac{0.25-p_0^2}{m} + p_0^2\right) & \alpha \in (-1, 0] \\ \frac{1}{4} & \alpha \in [0.5, 1) \\ \sum_{x \in I}\left(\frac{1-\alpha}{1-2\alpha}\right)^2\left(\frac{x}{m} - \frac{1}{2}\right)^2 p(x) & \\ \quad +\frac{1}{4} - \frac{1}{4}F_{m,p_0+\frac{1}{2}}\left(\frac{2-3\alpha}{2-2\alpha}m\right) & \\ \quad +\frac{1}{4}F_{m,p_0+\frac{1}{2}}\left(\frac{\alpha}{2-2\alpha}m\right) & \alpha \in (0, 0.5) \end{cases}$$

*where $I$ is the set of integers in $\left(\frac{\alpha m}{2-2\alpha}, \frac{(2-3\alpha)m}{2-2\alpha}\right]$, $F_{m,p_0+\frac{1}{2}}(x) := \sum_{k=0}^{\lfloor x \rfloor} p(x)$, and*

$$p(x) := \binom{m}{x}\left(\frac{1}{2} + p_0\right)^x\left(\frac{1}{2} - p_0\right)^{m-x}$$

*Proof of Lemma B.1.* Recall that $\hat{\theta}_0^*$ is given by

$$\theta_0^* = \begin{cases} \text{clip}\left(\frac{(1-|\alpha|)\bar{p}_0}{1-2\alpha}, -\frac{1}{2}, \frac{1}{2}\right), & 1 - 2\alpha > 0, \\ \text{sign}(\bar{p}_0)/2, & 1 - 2\alpha \leq 0. \end{cases}$$

We consider three cases for the value of $\alpha$:

(i) $\alpha \in (-1, 0]$

In this case we have

$$\theta_0^* = \frac{1-|\alpha|}{1-2\alpha}\bar{p}_0$$

and therefore

$$\mathbb{E}[\theta_0^*] = \frac{1-|\alpha|}{1-2\alpha}\bar{p}_0, \quad \mathbb{E}[(\theta_0^*)^2] = \left(\frac{1-|\alpha|}{1-2\alpha}\right)^2\mathbb{E}[\bar{p}_0^2] = \left(\frac{1-|\alpha|}{1-2\alpha}\right)^2\left(p_0^2 + \frac{\frac{1}{4} - p_0^2}{m}\right),$$

where we have used that $p_{0,i} \sim D_0$ for $i = 1, \ldots, m$, and thus $p_{0,i} + \frac{1}{2}$ follows a Bernoulli distribution with parameter $p_0 + \frac{1}{2}$.

(ii) $\alpha \in [0.5, 1)$

In this case, we have that

$$\theta_0^* = \begin{cases} \frac{1}{2} & \bar{p}_0 \geq 0 \\ -\frac{1}{2} & \bar{p}_0 < 0. \end{cases}$$

Since $\bar{p}_0 = \bar{q} - \frac{1}{2}$, where $\bar{q} := \frac{1}{m}\sum_{i=1}^m q_i$ and $q_i := p_{0,i}$, so that $q_i \sim Bern(p_0 + \frac{1}{2})$, we know that the events can be written as

$$\{\bar{p}_0 \geq 0\} = \{\bar{q} \geq 0.5\}, \quad \{\bar{p}_0 < 0\} = \{\bar{q} < 0.5\}.$$

Therefore,

$$\theta_0^* = \frac{1}{2}\chi_{\{\bar{q} \geq 0.5\}} - \frac{1}{2}\chi_{\{\bar{q} < 0.5\}}.$$

Finally, using the law of total expectation, we get that

$$\begin{aligned} \mathbb{E}[\theta_0^*] &= \mathbb{E}[\theta_0^*|\bar{q} \geq 0.5]\Pr[\bar{q} \geq 0.5] + \mathbb{E}[\theta_0^*|\bar{q} < 0.5]\Pr[\bar{q} < 0.5] \\ &= \frac{1}{2}\Pr[\bar{q} \geq 0.5] - \frac{1}{2}\Pr[\bar{q} < 0.5] \\ &= \frac{1}{2} - F_{m,p_0+\frac{1}{2}}(0.5m), \end{aligned}$$

where we have used that $m\bar{q} \sim Bin(m, p_0 + 0.5)$. Similarly for the second moment

$$
\begin{aligned}
\mathbb{E}[(\theta_0^*)^2] &= \mathbb{E}[(\theta_0^*)^2|\bar{q} \geq 0.5]\Pr[\bar{q} \geq 0.5] + \mathbb{E}[(\theta_0^*)^2|\bar{q} < 0.5]\Pr[\bar{q} < 0.5] \\
&= \frac{1}{4}\Pr[\bar{q} \geq 0.5] + \frac{1}{4}\Pr[\bar{q} < 0.5] \\
&= \frac{1}{4}.
\end{aligned}
$$

(iii) $\alpha \in (0, 0.5)$

In this case we have

$$
\theta_0^* = \begin{cases}
\frac{1-\alpha}{1-2\alpha}\bar{p}_0, & \text{if } \bar{p}_0 \in \left(-\frac{1-2\alpha}{2-2\alpha}, \frac{1-2\alpha}{2-2\alpha}\right] =: A \\
\frac{1}{2}, & \text{if } \bar{p}_0 > \frac{1-2\alpha}{2-2\alpha} =: B \\
-\frac{1}{2}, & \text{if } \bar{p}_0 \leq -\frac{1-2\alpha}{2-2\alpha} =: C
\end{cases}
$$

where we have denoted by $A, B, C$ the random events that we have not clipped the value of the performative estimator, that we have clipped it from above or that we have clipped in from below. Using the law of total expectation, we have

$$
\begin{aligned}
\mathbb{E}[\theta_0^*] &= \mathbb{E}[\theta_0^*|A]\Pr[A] + \mathbb{E}[\theta_0^*|B]\Pr[B] + \mathbb{E}[\theta_0^*|C]\Pr[C] \\
&= \mathbb{E}[\theta_0^*\chi_A] + \frac{1}{2}\Pr[B] - \frac{1}{2}\Pr[C] \\
&= \mathbb{E}[\theta_0^*\chi_A] + \frac{1}{2}\Pr\left[\bar{q} > \frac{2-3\alpha}{2-2\alpha}\right] - \frac{1}{2}\Pr\left[\bar{q} \leq \frac{\alpha}{2-2\alpha}\right]
\end{aligned}
$$

The first term can be computed as follows

$$
\mathbb{E}[\theta_0^*\chi_A] = \sum_{x \in I} \frac{1-\alpha}{1-2\alpha}\left(\frac{x}{m} - \frac{1}{2}\right)p(x),
$$

where we have used that $m\bar{p}_0 + m/2 \sim Bin(m, p_0 + \frac{1}{2})$ and have denoted by $p(x)$ the PMF of $Bin(m, p_0 + \frac{1}{2})$. The last two terms are easily expressed via the CDF of the same distribution, giving us that

$$
\mathbb{E}[\theta_0^*] = \sum_{x \in I}\left(\frac{1-\alpha}{1-2\alpha}\right)\left(\frac{x}{m} - \frac{1}{2}\right)p(x) + \frac{1}{2} - \frac{1}{2}F_{m,p_0+\frac{1}{2}}\left(\frac{2-3\alpha}{2-2\alpha}m\right) - \frac{1}{2}F_{m,p_0+\frac{1}{2}}\left(\frac{\alpha}{2-2\alpha}m\right).
$$

where $I$ is the set of integers in the interval $(\frac{\alpha}{2-2\alpha}m, \frac{2-3\alpha}{2-2\alpha}m]$. Similarly, for the second moment we have that

$$
\begin{aligned}
\mathbb{E}[(\theta_0^*)^2] &= \mathbb{E}[(\theta_0^*)^2|A]\Pr[A] + \mathbb{E}[(\theta_0^*)^2|B]\Pr[B] + \mathbb{E}[(\theta_0^*)^2|C]\Pr[C] \\
&= \mathbb{E}[(\theta_0^*)^2\chi_A] + \frac{1}{4}\Pr[B] + \frac{1}{4}\Pr[C] \\
&= \mathbb{E}[(\theta_0^*)^2\chi_A] + \frac{1}{4}\Pr\left[\bar{q} > \frac{2-3\alpha}{2-2\alpha}\right] - \frac{1}{2}\Pr\left[\bar{q} \leq \frac{\alpha}{2-2\alpha}\right] \\
&= \sum_{x \in I}\left(\frac{1-\alpha}{1-2\alpha}\right)^2\left(\frac{x}{m} - \frac{1}{2}\right)^2 p(x) + \frac{1}{4} - \frac{1}{4}F_{m,p_0+\frac{1}{2}}\left(\frac{2-3\alpha}{2-2\alpha}m\right) + \frac{1}{4}F_{m,p_0+\frac{1}{2}}\left(\frac{\alpha}{2-2\alpha}m\right),
\end{aligned}
$$

which finishes the proof.

$\square$

Now, we are ready to present the full proof.

*Proof of Theorems 4.3 and A.2.* We begin by rewriting the expected loss as follows

$$
\begin{aligned}
\mathbb{E}[(\theta_0 - z_0)^2] &= \mathbb{E}[\mathbb{E}[\theta_0^2 - 2\theta_0 z_0 + z_0^2 | \theta_0]] \\
&= \mathbb{E}[\theta_0^2 - 2\theta_0 \, \mathbb{E}[z_0 | \theta_0] + \mathbb{E}[z_0^2 | \theta_0]] \\
&= \mathbb{E}\left[\theta_0^2 - 2\theta_0 p_1(\theta_0) + \frac{1}{4}\right] \\
&= (1 - 2\alpha)\, \mathbb{E}[\theta_0^2] - 2(1 - |\alpha|) p_0 \, \mathbb{E}[\theta_0] + \frac{1}{4}
\end{aligned}
$$

where the expectation is only in terms of the randomness of the observations $\{p_{0,i}\}_{i=1}^m$.

For the naive estimator, $\hat{\theta}_0^n$, we have that the first two moments are

$$
\mathbb{E}[\hat{\theta}_0^n] = p_0
$$

$$
\mathbb{E}[(\hat{\theta}_0^n)^2] = p_0^2 + \frac{(\frac{1}{2} - p_0)(\frac{1}{2} + p_0)}{m},
$$

which follows since $p_{0,i} \sim D_0$ for $i = 1, \ldots, m$. Therefore, we get

$$
\begin{aligned}
\mathbb{E}[(\hat{\theta}_0^n - z)^2] &= (1 - 2\alpha)\, \mathbb{E}[\theta_0^2] - 2(1 - |\alpha|) p_0 \, \mathbb{E}[\theta_0] + \frac{1}{4} \\
&= p_0^2(2|\alpha| - 2\alpha - 1) + \frac{1}{4} + \frac{(2\alpha - 1)(4p_0^2 - 1)}{4m}
\end{aligned}
$$

For the performative estimator, we use the first and second moments of $\hat{\theta}_0^*$ from Lemma B.1 to obtain

$$
\mathbb{E}[(\hat{\theta}_0^* - z)^2] =
$$
$$
\begin{cases}
\frac{(1-|\alpha|)^2}{1-2\alpha}\left(\frac{\frac{1}{4} - p_0^2}{m} - p_0^2\right) + \frac{1}{4} & \alpha \in (-1, 0] \\[2mm]
p_0(1 - |\alpha|)\left(2 F_{m, p_0 + \frac{1}{2}}\left(\frac{m}{2}\right) - 1\right) + \frac{1-\alpha}{2} & \alpha \in [0.5, 1) \\[2mm]
\sum_{x \in I}\left((1 - 2\alpha)g(x)^2 - 2(1 - |\alpha|)p_0 g(x)\right)p(x) + \left(p_0(1 - |\alpha|) - \frac{1-2\alpha}{4}\right) F_{m, p_0 + \frac{1}{2}}\left(\frac{2-3\alpha}{2-2\alpha}m\right) & \\[2mm]
+ \left(p_0(1 - |\alpha|) + \frac{1-2\alpha}{4}\right) F_{m, p_0 + \frac{1}{2}}\left(\frac{\alpha m}{2-2\alpha}\right) - p_0(1 - |\alpha|) + \frac{1-\alpha}{2}, & \alpha \in (0, 0.5),
\end{cases}
$$

where $g(x) := \left(\frac{1-\alpha}{1-2\alpha}\right)\left(\frac{x}{m} - \frac{1}{2}\right)$.

Asymptotically, as $m \to \infty$, we have that the moments of $\hat{\theta}_0^*$ for $\alpha \in (-1, 0]$ are given by

$$
\mathbb{E}[\hat{\theta}_0^*] = \frac{(1 - |\alpha|)}{1 - 2\alpha} p_0 \to \frac{(1 - |\alpha|)}{1 - 2\alpha} p_0
$$

$$
\mathbb{E}[(\hat{\theta}_0^*)^2] = \frac{(1 - |\alpha|)^2}{(1 - 2\alpha)^2}\left(\frac{0.25 - p_0^2}{m} + p_0^2\right) \to \frac{(1 - |\alpha|)^2}{(1 - 2\alpha)^2} p_0^2
$$

Similarly, for $\alpha \in [0, 5, 1)$, we have

$$
\mathbb{E}[\hat{\theta}_0^*] = \frac{1}{2} - F_{m, p_0 + \frac{1}{2}}\left(\frac{m}{2}\right) \to \frac{sign(p_0)}{2}
$$

$$
\mathbb{E}[(\hat{\theta}_0^*)^2] = \frac{1}{4} \to \frac{1}{4}
$$

where we have used that the CDF function $F_{m, p_0 + \frac{1}{2}}\left(\frac{m}{2}\right)$ converges to 1 for non-negative $p_0$ and to 0 for negative $p_0$ as $m \to \infty$.

Finally, for $\alpha \in (0, 0.5)$, we have that

$$
\begin{aligned}
\mathbb{E}[\hat{\theta}_0^*] &= \mathbb{E}\left[ \text{clip}\left( \frac{1 - |\alpha|p_0}{1 - 2\alpha}, -\frac{1}{2}, \frac{1}{2} \right) \right] \\
&= \mathbb{E}\left[ \frac{(1 - |\alpha|)\bar{p}_0}{1 - 2\alpha} \chi_{\{\bar{p}_0 \in A\}} \right] + \frac{1}{2} \Pr[\bar{p}_0 \in B] - \frac{1}{2} \Pr[\bar{p}_0 \in C] \\
&\to \frac{(1 - |\alpha|)p_0}{1 - 2\alpha} \chi_{\{p_0 \in A\}} + \frac{1}{2} \chi_{[p_0 \in B]} - \frac{1}{2} \chi_{[p_0 \in C]} \\
&= \mathbb{E}[\theta_0^* \mid \alpha \in (0, 0.5)].
\end{aligned}
$$

where $A$ denotes the region (a function of $\alpha$), where $\hat{\theta}_0^*$ has not been clipped, $B$ represents the region where it has been clipped from above, and $C$ is the region where it has been clipped from below. The third line follows from: (1) the law of large numbers, which ensures that $\bar{p}_0 \to p_0$ almost surely as $m \to \infty$, and (2) the dominated convergence theorem. The same argument applies for $\mathbb{E}[(\hat{\theta}_0^*)^2]$. Thus, combining this with the other two cases for $\alpha$, we get the following asymptotic results

$$
\lim_{m \to \infty} \mathbb{E}[\hat{\theta}_0^*] = \theta_0^*, \quad \lim_{m \to \infty} \mathbb{E}[(\hat{\theta}_0^*)^2] = (\theta_0^*)^2.
$$

Therefore, we can conclude that as $m \to \infty$,

$$
\mathbb{E}[(\hat{\theta}_2^* - z)^2] \to \texttt{loss}_0^*.
$$

$\square$

### B.5. Proof of Theorem 5.1

Consider Lagrangian function

$$
\begin{aligned}
L(\boldsymbol{w}, \boldsymbol{q}, \boldsymbol{\nu}, \boldsymbol{\mu}, \boldsymbol{\eta}) :=& \\
&\sum_{t=0}^{\infty} \gamma^t(\theta_t^2 - 2\theta_t p_{t+1}) - (\alpha\theta_t + \beta p_t + (1 - |\alpha|)(1 - \lambda)\pi - p_{t+1})\nu_t - (1/2 - \theta_t)\mu_t - (\theta_t + 1/2)\eta_t.
\end{aligned}
$$

KKT conditions for this infinite-horizon problem (see Section 4.5 of Stokey et al., 1989) give

$$
\begin{aligned}
0 &= 2\gamma^t(\theta_t - p_{t+1}) - \alpha\nu_t + \mu_t - \eta_t, \\
0 &= -2\gamma^t \theta_t + \nu_t - \beta\nu_{t+1}, \\
0 &= (1/2 - \theta_t)\mu_t, \mu_t \geq 0, \\
0 &= (\theta_t + 1/2)\eta_t, \eta_t \geq 0.
\end{aligned}
$$

Thus, the solution for the case when the restrictions on $\theta_t$ are non-binding satisfies

$$
\begin{aligned}
\theta_{t+1} &= \frac{(1 - 2\alpha + \gamma\alpha\beta^2)}{\gamma(1 - \alpha)\beta}\theta_t - \frac{1 - \gamma\beta^2}{\gamma(1 - \alpha)\lambda}s_{t+1} + \frac{\beta(1 - \lambda)}{(1 - \alpha)\lambda}\pi, \\
s_{t+2} &= \alpha\lambda\theta_t + \beta s_{t+1} + (1 - \lambda)\pi.
\end{aligned}
$$

We get that the optimal path satisfies a first-order linear recurrence relation for $\theta_t$ and $s_t$. Its characteristic equation follows

$$
x^2 - \frac{1 - 2\alpha + \gamma\beta^2}{\gamma(1 - \alpha)\beta}x + \frac{1}{\gamma} = 0.
$$

It gives the following eigenvalues

$$
x_{0,1} = \frac{1 - 2\alpha + \gamma\beta^2 \pm \sqrt{(1 - \gamma\beta^2)((1 - 2\alpha)^2 - \gamma\beta^2)}}{2\gamma(1 - \alpha)\beta}.
$$

Notice that the product of these eigenvalues is $1/\gamma$. Thus, one of the eigenvalues is necessarily bigger than 1 in absolute value. Due to the restrictions on $w$, the homogeneous solution corresponding to this eigenvalue should be zero.

Consider the case of $1 - 2\alpha \geq \sqrt{\gamma}\beta$, then, the smallest eigenvalue, $\omega$, satisfies

$$\omega = \beta + \frac{(1 - 2\alpha)(1 - \gamma\beta^2) - \sqrt{(1 - \gamma\beta^2)((1 - 2\alpha)^2 - \gamma\beta^2)}}{2\gamma(1 - \alpha)\beta} = \beta + \frac{2\alpha\beta}{1 - 2\alpha + \xi}.$$

Corresponding eigenvector gives the following homogeneous solution

$$s_{t+1}^h = s\omega^t, \ \theta_t^h = \frac{2(1 - |\alpha|)}{1 - 2\alpha + \xi} r\omega^t.$$

One of inhomogeneous solutions satisfies

$$s_{t+1}^i = \frac{(1 - 2\alpha - \gamma\beta(1 - \alpha))(1 - \lambda)\pi}{1 - 2\alpha - \beta + \alpha\beta - \gamma\beta(1 - \alpha - \beta)}, \ \theta_t^i = \frac{(1 - \gamma\beta)(1 - |\alpha|)(1 - \lambda)\pi}{1 - 2\alpha - \beta + \alpha\beta - \gamma\beta(1 - \alpha - \beta)}.$$

Using the initial conditions, we get the desired solution.

### B.6. Proof of Theorem 5.2

Consider Lagrangian function

$$L(\boldsymbol{w}, \boldsymbol{q}, \boldsymbol{\nu}, \boldsymbol{\mu}, \boldsymbol{\eta}) :=$$
$$\sum_{t=0}^{\infty} \gamma^t(\theta_t^2 - 2\theta_t p_t) - (\alpha\theta_t + \beta p_t + (1 - |\alpha|)(1 - \lambda)\pi - p_{t+1})\nu_t - (1/2 - \theta_t)\mu_t - (\theta_t + 1/2)\eta_t.$$

KKT conditions for this infinite-horizon problem (see Section 4.5 of Stokey et al., 1989) give

$$0 = 2\gamma^t(\theta_t - p_t) - \alpha\nu_t + \mu_t - \eta_t,$$
$$0 = -2\gamma^t\theta_t + \nu_{t-1} - \beta\nu_t,$$
$$0 = (1/2 - \theta_t)\mu_t, \mu_t \geq 0,$$
$$0 = (\theta_t + 1/2)\eta_t, \eta_t \geq 0.$$

Thus, the solution for the case when the restrictions on $\theta_t$ are non-binding satisfies

$$\theta_{t+1} = \frac{1 + \gamma\alpha\beta}{\gamma(\alpha + \beta)}\theta_t - \frac{1 - \gamma\beta^2}{\gamma(\alpha + \beta)}p_t + \frac{\beta(1 - |\alpha|)(1 - \lambda)}{\alpha + \beta}\pi,$$
$$p_{t+1} = \alpha\theta_t + \beta p_t + (1 - |\alpha|)(1 - \lambda)\pi.$$

We get that the optimal path satisfies a first-order linear recurrence relation for $\theta_t$ and $p_t$. Its characteristic equation follows

$$x^2 - \frac{1 + \gamma\beta(2\alpha + \beta)}{\gamma(\alpha + \beta)}x + \frac{1}{\gamma} = 0.$$

It gives the following eigenvalues

$$x_{0,1} = \frac{1 + \gamma\beta(2\alpha + \beta) \pm \sqrt{(1 - \gamma\beta^2)(1 - \gamma(2\alpha + \beta)^2)}}{2\gamma(\alpha + \beta)}.$$

Notice that the product of these eigenvalues is $1/\gamma$. Thus, one of the eigenvalues is necessarily bigger than 1 in absolute value. Due to the restrictions on $w$, the homogeneous solution corresponding to this eigenvalue should be zero.

The smallest eigenvalue, $\kappa$, satisfies

$$\kappa = \frac{1 + \gamma\beta(2\alpha + \beta) - \sqrt{(1 - \gamma\beta^2)(1 - \gamma(2\alpha + \beta)^2)}}{2\gamma(\alpha + \beta)} = \beta + \frac{2\alpha}{1 + \chi}.$$

Thus, homogeneous solution follows

$$q_t^h = q\kappa^t, \ \theta_t^h = \frac{2}{1+\chi}q\kappa^t.$$

One of inhomogeneous solutions satisfies

$$q_t^i = \frac{(1-\gamma(\alpha+\beta))(1-|\alpha|)(1-\lambda)\pi}{1-\alpha-\beta-\gamma(\alpha+\beta-\beta(2\alpha+\beta))}, \ \theta_t^i = \frac{(1-\gamma\beta)(1-|\alpha|)(1-\lambda)\pi}{1-\alpha-\beta-\gamma(\alpha+\beta-\beta(2\alpha+\beta))}.$$

### B.7. Proof of Proposition A.1

Using the results of Proposition 4.1, we get

$$\theta_1^* = \begin{cases} \mathrm{clip}\left(\frac{(1-|\alpha|)s_2}{1-2\alpha}, -\frac{1}{2}, \frac{1}{2}\right), & 1-2\alpha > 0, \\ \frac{\mathrm{sign}(s_2)}{2}, & 1-2\alpha \le 0, \end{cases}$$

which results in the following loss in the second period:

$$(\theta_1^*)^2 - 2\theta_1^* p_2^* = \begin{cases} -\frac{(1-|\alpha|)^2 s_2^2}{1-2\alpha}, & 2(1-|\alpha|)|s_2| < 1-2\alpha, \\ \frac{1-2\alpha}{4} - (1-|\alpha|)|s_2|, & 2(1-|\alpha|)|s_2| \ge 1-2\alpha. \end{cases}$$

Notice that

$$(\theta_1^*)^2 - 2\theta_1^* p_2^* \ge -\frac{(1-|\alpha|)^2 s_2^2}{1-2\alpha}.$$

Thus,

$$\sum_{t=0}^{1} \gamma^t(\theta_t^2 - 2\theta_t p_{t+1}) \ge \theta_0^2 - 2\theta_0 p_1 - \frac{\gamma(1-|\alpha|)^2 s_2^2}{1-2\alpha}.$$

So, if the minimizer of the right hand side satisfies $2(1-|\alpha|)|s_2^{\mathrm{rhs},*}| \le 1-2\alpha$, it will minimize the left-hand side.

Similarly to Appendix B.2, we have that the minimizer of the right-hand side satisfies

$$\theta_0^{\mathrm{rhs},*} =$$
$$\begin{cases} \mathrm{clip}\left(\frac{(1-|\alpha|)((1-2\alpha+\gamma\alpha\beta^2)s_1 + \gamma\alpha\beta(1-\lambda)\pi)}{(1-2\alpha)^2 - \gamma\alpha^2\beta^2}, -\frac{1}{2}, \frac{1}{2}\right), & 1-2\alpha > \sqrt{\gamma}|\alpha|\beta, \\ \frac{\mathrm{sign}((1-2\alpha+\gamma\alpha\beta^2)s_1 + \gamma\alpha\beta(1-\lambda)\pi)}{2}, & 1-2\alpha \le \sqrt{\gamma}|\alpha|\beta. \end{cases}$$

Thus, when $1-2\alpha > \sqrt{\gamma}|\alpha|\beta$ and $2(1-|\alpha|)|s_2^*| \le 1-2\alpha$, we get the desired solution to our problem.

# C. Details of RL Simulations

This section gives additional details about RL-like simulations in Sections 4.2.2 and 5.2.

## C.1. One-period Episodic Simulations

We consider episodic exploration of $T = 1$ slow model, where we additionally assume that $\lambda = 0$ and $\lambda$ is known to the provider. In this setting, we assume that each episode has the following structure.

1. Nature samples $q_0$.

2. The provider observes $\{z_0^i\}_{i=0}^{m-1} \sim D_0^m$.

3. The provider deploys $\theta_0$.

4. The provider observes $\{z_1^i\}_{i=0}^{m-1} \sim D_1^m$.

We implement Algorithm 1, adopted version of Algorithm 1 of Liu et al. (2022), where we denote the episode number by $\tau$, with hyperparameter $\beta = 2^{-8}$ to find the optimal prediction. (Notice that the first period observations are non-informative for the log-likelihood maximization because $\lambda = 0$.)

---

**Algorithm 1** Optimistic Maximum Likelihood Estimation

---

  **Initialize:** $B^0 = \{(\alpha, \pi) : \alpha \in [-1, 1], \pi \in [-1/2, 1/2]\}, \mathcal{D} = \{\}$
  **for** $\tau = 0$ **to** $T$ **do**
    Deploy $\theta_0^\tau = \arg\min_{\theta \in [-1/2, 1/2]} \min_{(\alpha,\pi) \in B^\tau} \texttt{loss}(\theta \mid \alpha, \pi)$
    Observe $S_1^\tau \sim (D_1^\tau)^m$
    Add $(\theta_0^\tau, S_1^\tau)$ to $\mathcal{D}$
    Update $B^{\tau+1} = \left\{ (\alpha, \pi) \in B^0 : \sum_{(\theta, S) \in D} \log \Pr(S \mid \alpha, \pi, \theta) \geq \max_{(\alpha,\pi) \in B^0} \sum_{(\theta, S) \in D} \log \Pr(S \mid \alpha, \pi, \theta) - \beta \right\}$
  **end for**

---

## C.2. Infinite Horizon Simulations

We consider episodic exploration of $T = \infty$ slow model, where the provider know the value of $\gamma$. In this setting, we assume that each step has the following structure.

1. The provider observes $\{z_t^i\}_{i=0}^{m-1} \sim D_t^m$.

2. The provider deploys $\theta_t$.

For this case, we implement heuristic Algorithm 2, where we denoted the value function as

$$V(p_0, \alpha, \pi, \lambda, \gamma) := \min_{(\theta_t)_{t=0}^{\infty}} \sum_{t=0}^{\infty} \gamma^t \texttt{loss}(\theta_t \mid \theta_{t-1}, \ldots, \theta_0, p_0, \alpha, \pi, \lambda).$$

This algorithm learns the performative response by deploying extreme predictions $\{-1/2, 1/2\}$ at random for the first $4$ steps. Then the model provider learns the parameters of the performative response by likelihood maximization and deploys the optimal policy under their estimates.

---

**Algorithm 2** Greedy Exploration

---

Observe $S_0 \sim D_0^m$
**for** $t = 0$ **to** $3$ **do**
   Deploy $\theta_t$ at random from $\{-1/2, 1/2\}$
   Observe $S_{t+1} \sim D_{t+1}^m$
**end for**
**for** $t = 4$ **to** $T$ **do**
   Estimate $(\alpha, \pi, \lambda, p_0) = \arg\max_{\alpha, \pi, \lambda, p_0} \sum_{\tau=0}^{t} \log \Pr(S_t \mid \theta_{t-1}, \ldots, \theta_0, p_0, \alpha, \pi, \lambda)$
   Deploy $\theta_t = \arg\min_{\theta_t} \texttt{loss}(\theta_t \mid p_{t+1}(\theta_t, \ldots, \theta_0, p_0, \alpha, \pi, \lambda)) + \gamma V(p_{t+1}(\theta_t, \ldots, \theta_0, p_0, \alpha, \pi, \lambda), \alpha, \pi, \lambda, \gamma)$
   Observe $S_{t+1} \sim D_{t+1}^m$
**end for**

---

