# OpenReview forum: "On the Impact of Performative Risk Minimization for Binary Random Variables"
_ICML.cc/2025/Conference — ICML 2025 poster_

### Official Review · Reviewer_y6Ux · 2025-03-10

**Overall Recommendation:** 2

**Summary:**

The paper devises two metrics to study the path taken for models in a performative setting and applies them in the study of binary random variables (shifted Bernoulli).

**Claims And Evidence:**

-   They claim that they analyze the impact of Performative Risk Minimization. However, as far as I can tell, there is no such thing in the previous literature (especially not in Perdomo et al. (2020)) and the paper never defines what they understand by PRM. They cite three papers but those solve different optimization problems: Perdomo et al. (2020) and Brown et al. (2022) optimize for the stable point, while Ray et al. (2020) optimizes for the optimal point. It is unclear in the beginning of the paper what they are trying to do and what is their objective. In addition, they claim that Performative Prediction tries to minimize the objective function presented in (1). However, the standard definition of Performative Prediction in Perdomo et al. (2020) does not fall into this equation. It is unclear how the other papers relate to equation (1) too.

-   The usefulness of the metrics cannot be fully supported with just the example of binary random variables. I would suggest extending the analysis for other problems (like what other papers in the field – like Izzo et al. (2022) – do). Furthermore, there is lack of motivation of the binary problem; why is it relevant for Performative Prediction? A slight justification only appears in the last section with the use cases. The case studies are too simplistic to be considered realistic and, therefore, it is hard to understand why this is an interesting problem for Performative Prediction.

**Essential References Not Discussed:**

- The paper doesn't discuss Lin & Zrinic (2024), Izzo et al. (2022)  and *Performative Prediction with Neural Networks* (Mofakhami et al. 2023) when saying  “To our awareness, the only work that studies properties beyond accuracy is that of Jin et al. (2024)”. Works in Performative Prediction study the Performative Loss generally, which sometimes might not transfer into accuracy (see *Performative Prediction in NN*). Nevertheless, others like Izzo et al. (2022) or Lin & Zrinic (2024) study how the model converges to the closed-formed solution. This could be fixed if accuracy was changed by “performance”.
- The paper doesn't discuss how the regret proposed in *Regret minimization with Performative Feedback* (Jaagadeesan et al. 2022) is similar to the mean shift that you propose in your paper. Its citation is also missing.
- The paper lacks a discussion of how your equation (1) is similar to the discounted loss that they consider in *Performative Reinforcement Learning* (Mandal et al. 2022). Its citation is also missing.

**Experimental Designs Or Analyses:**

-   I did not reproduce the results of experimental experiments

**Methods And Evaluation Criteria:**

-   The claims on the binary random variables are supported mainly by theoretical evidence, which make sense in this case.

**Other Comments Or Suggestions:**

none

**Other Strengths And Weaknesses:**

-   Strengths

    -   The paper identifies and explores the impacts of Performative Prediction with a broader scope that just performance of the models, which is an very interesting direction for the Performative Prediction field.

-   Weaknesses

    -   The problem studied is too narrow. Would it be possible to apply the metrics to other problems? How would the conclusions be different? Why is the binary random variables problem key for Performative Prediction?

    -   Conclusions are too shallow for the number of theoretical derivations, experiments and plots. (“Our results suggest that the performatively optimal (PRM) path is, in general, biased and introduces a non-zero mean shift”). Isn’t this conclusion expected, as there is a distribution shift? Why are all mathematical derivations relevant?

    -   Relation to the previous literature is not clear enough (See claims and evidence, first point)

**Questions For Authors:**

See weaknesses.

I would be willing to change my evaluation of the paper if the metrics were applied to other problems or if it is better justified why the binary random variables problem is key for Performative Prediction. Furthermore, authors would need to show if these metrics can be used to obtain deeper conclusions. Another direction to improve my evaluation can be to study more realistic case studies where the effects of biased paths can be shown.

**Relation To Broader Scientific Literature:**

-   This paper adds a couple of metrics to study the bias of estimators in Performative Prediction.

-   The link of the optimization algorithm they use “performative risk minimization” and the loss is not very clear

-   The paper seems like an extension of examples used before in papers like Perdomo et al. (2020) or Izzo. et al (2022). Nevertheless, in these papers they used them to exemplify their claims, not as their main results.

**Theoretical Claims:**

-   I have not checked the correctness of proofs. The assumptions (loss and problem) of the problem should be better explained (see **Claims and Evidence**)

---

> ### Author Rebuttal · Authors · 2025-04-01
>
> Thank you for the detailed and constructive review! We address your concerns and questions below and remain available for further discussion.
>
> **The paper never defines what they understand by PRM … They cite three papers but those solve different optimization problems**
>
> Thank you for your comment. The optimization problem (1), whose solution we analyze, in the special case of T = 1 and slow deployment, is exactly the classic problem of finding a performatively-optimal model, introduced in Perdomo et al. The cases of T > 1 and rapid deployment cover extensions discussed in Brown et al. and Ray et al..
>
> By Performative Risk Minimization we simply mean the solution (sequence of models) which minimizes the performative risk. This is exactly the solution to problem (1), which we introduce at the start of Section 3. It is also equivalent to finding a performatively-optimal path.
>
> While many works develop methods for finding performatively-stable models instead, this is arguably because of the hardness of finding performatively-optimal points in general settings. Already Perdomo et al. argue (see Section 2 in their ICML version) that minimizing the performative risk is a natural objective in performative prediction. They also show that, under some assumptions, stable and optimal points are “nearby” (their Section 4).
>
> **The usefulness of the metrics cannot be fully supported with just the example of binary random variables … The paper seems like an extension of examples used before in papers like Perdomo et al. (2020) or Izzo. et al (2022).”**
>
> The studied learning problem is indeed less general than those usually considered in the performative prediction literature. We refer to our response to reviewer 5zXv for an explanation of how our work can serve as a foundation for analyzing more general learning tasks.
>
> However, we also study a different mathematical problem, compared to that prior work.
>
> Works in performative prediction focus on finding performatively optimal/stable points. In contrast, we study the properties (e.g. bias and shift) of models that minimize the performative risk (assuming that such are already found). This requires a different set of mathematical tools, as existing assumptions (e.g.  convexity, smoothness, Lipschitzness w.r.t the Wasserstein distance) from prior work are suited for optimizers searching for optimal/stable points and do not necessarily facilitate an analysis of our problem.
>
> We restrict to the analysis of a mean estimation problem for two reasons:
>
> - First, to understand the impact of choosing performatively optimal models on the distribution, we need structural assumptions on how the distribution changes as a function of the deployed model. To our awareness, such assumptions have previously only been proposed in specific examples (e.g. those referred to by the reviewer, location families etc). Creating plausible models of how general distributions change in performative settings is indeed an interesting research direction, but orthogonal to our work.
>
> - Second, obtaining theoretically-precise results about the impact of PRM was already challenging for binary mean estimation, as is evident by the need to resort to simulations in more advanced cases. We certainly agree that exploring other learning setups in the future is an important research direction, but we expect that this will have to be limited to simulations, due to likely mathematical intractability. While we hope that our work can serve as a useful foundation for such exploration, this is beyond the scope of our theoretical paper. This is also clearly reflected in the paper title and the contributions part of the introduction.
>
> **Related work**
>
> Thank you for the suggested papers. We will cite and discuss them.
>
> We agree that minimizing the performative loss does not always translate to accuracy. We indeed intended to say that other works focus on loss minimization. We will edit the text as suggested and cite the works in the related work section.
>
> We will also cite Jaagadeesan et al. 2022 and Mandal et al. 2022, alongside with Brown et al. (2022) and Ray et al. (2022a). All these works indeed motivate our loss (1), whose minimizers we study throughout the paper.
>
> We do not see a link between our mean shift metric and the regret of Jaagadeesan et al. Could the reviewer clarify?
>
> **Would it be possible to apply the metrics to other problems?**
>
> We again refer to our response to reviewer 5zXv for a thorough discussion.
>
> **Isn’t this conclusion expected, as there is a distribution shift? Why are all mathematical derivations relevant?**
>
> Our theory not only proves the presence of, but also quantifies the shift and bias, for a wide range of problem parameters. The reason we put such a general comment as a summary was that conclusions about the direction and amount of shift and bias are, in general, case-dependent. We have described the more detailed, case-specific conclusions throughout the technical sections.

---

> > ### Comment · Reviewer_y6Ux · 2025-04-04
> >
> > Thank you for the clarifications about the concepts you use and the problem you are dealing with. While I see how the problem is defined in the beginning of Section 3, I’d suggest to include them before. Clearly stating in the introduction the difference between stable and optimal points and link them to RM and PRM, respectively, would help a lot. Furthermore, improving the clarity of the related work section would also help. Explicitly stating what kind of problems past works have addressed when finding the *optimal* point would give a better context and motivation for the problem (what you say in the related work section of this rebuttal might actually be enough - “We indeed intended to say that other works focus on loss minimization”).
> >
> > Explaining better the impact and restrictions of a mean estimation problem was also helpful to understand why you consider a narrow problem. I think it’d be important to include them in the main version of the text. While I consider *very* useful to initiate the discussions of broader effects of PRM, the focus on bias and mean shift seems somehow straight forward. I have doubts of how direct can all other tools developed by the paper be applied into the simulations that you propose for more advanced settings. Could you provide more explicit details of how your paper can be used as foundation for that future research? The contribution and impact of the paper wouldn’t be as limited if other settings were considered (or at least a more direct and explicit connection was made).
> >
> > Taking into account that all my previous comments could be included in the main text and the limitations discussed, I will raise my score to 2.

---

> > > ### Author Response · Authors · 2025-04-04
> > >
> > > Thank you for your response and for engaging in further discussion! We are glad to hear that our clarifications have been helpful.
> > >
> > > We thank the reviewer for their suggestions, which we agree will greatly benefit the exposition. As suggested, we will include the above discussions in the next version of the manuscript. In particular, we will clarify the relation between stable and optimal points and link them to RM and PRM early on in the text. We will also clarify the statement in the related work section and expand on the relevance and limitations of the mean estimation setup in the main text.
> > >
> > > To further explain how our paper can serve as a foundation for future work, we note that the notions we consider in the binary setting have analogs in general learning tasks as well. In particular, we discuss below how the notions of mean shift, bias, and the error-uncertainty trade-off can be generalized to more complex settings. Therefore, the notions we have introduced in our work, together with the general framework of Section 3.1, can serve as a foundation for future work on exploring the impact of PRM.
> > >
> > > Regarding the **mean shift metric**, we identify two possible extensions, for parametric and non-parametric distributions, respectively.
> > >
> > > * Consider a setting where the distribution is parametrized by a finite number of parameters, $D_t = D(\mathbf{w}_t)$, where $\mathbf{w}_t = (w^1_t, \dots, w^k_t)$. This is, for example, the case for distributions defined via causal graphical models, as well as for many common distributions (e.g., exponential families). Then, one can define the parameter shift metric as $\mathtt{shift}_t = E(\mathbf{w}_t - \mathbf{w}^0_t)$, where $\mathbf{w}^0_t$ are the parameters of the distribution at time $t$ if the distribution was not affected by the performativity.
> > >
> > > * In a non-parametric setting, we can instead define a divergence-based metric $\mathtt{shift}_t = E(K(D_t, D^0_t))$, where $K$ is an arbitrary divergence function (e.g., KL-divergence). The function $K$ can be designed to capture an undesirable shift in the distribution, according to the target application.
> > >
> > > These metrics can then be studied under different models for the performative response and counterfactual dynamics of the distribution in the absence of performativity, which can be chosen depending on the learning task and application under consideration.
> > >
> > > Regarding the **bias metric**, we provide two extensions suitable for cases where the distribution is divided into several groups, which is relevant, e.g., in fairness-sensitive applications.
> > >
> > > * First, in a setting with several groups and multi-labeled data, one could calculate **a matrix of biases with one entry for each group and label** defined as follows $\mathtt{bias}^{g,y} = E(E_{(X, Y, G) \sim D^{test}_t}(q^y_t(X) - [Y=y] | G=g))$, where $q_t(X) = (q^1_t(X), \dots, q^{|\mathcal{Y}|}_t(X))$ is the vector of model's softmax probabilities at time $t$ and $G$ is the group. These biases could be interpreted as a measure of unfairness among groups.
> > >
> > > * Second, one can use established metrics from the literature of **bias amplification** (see [1,2,3,4]).
> > >
> > > Finally, regarding the **error-uncertainty** trade-off, consider an arbitrary supervised learning task, where the goal is to find predictors $h_t \colon \mathcal{X}\to\mathcal{Y}$ from a set of possible models $\mathcal{H}$ (e.g. linear models or neural networks of a certain architecture). Suppose that we deploy a model $h_{t-1}$, which induces a distribution $D_t$. Then, the performative loss of $h_{t-1}$ in the slow deployment is $E_{(X, Y) \sim D_t}(l(h_{t-1}(X), Y)) = (E_{(X, Y) \sim D_t}(l(h_{t-1}(X), Y)) - \min_{h’ \in \mathcal{H}} E_{(X, Y) \sim D_t}(l(h'(X), Y))) +\min_{h’ \in \mathcal{H}} E_{(X, Y) \sim D_t}(l(h'(X), Y))$
> > >
> > > The first term measures how far from optimal the model is on its own distribution. The second term corresponds to the simplicity of the induced distribution, as measured by how well the model class can fit it. We thus observe an error-simplicity trade-off, akin to the trade-off observed in Lemma 3.2. Moreover, as discussed in Section 3, the decomposition in Lemma 3.2 holds for a large class of distributions [5].
> > >
> > > We will include the discussion of these extensions in the next version of the manuscript.
> > >
> > > **References**
> > >
> > > [1] Jieyu Zhao, Tianlu Wang, Mark Yatskar, Vicente Ordonez, and Kai-Wei Chang. Men also like shopping: Reducing gender bias amplification using corpus-level constraints. arXiv preprint arXiv:1707.09457, 2017.
> > >
> > > [2] Angelina Wang and Olga Russakovsky. Directional bias amplification. ICML, 2021.
> > >
> > > [3] Dora Zhao, Jerone Andrews, and Alice Xiang. Men also do laundry: Multi-attribute bias amplification. ICML, 2023.
> > >
> > > [4] Bhanu Tokas, Rahul Nair, Hannah Kerner. Making Bias Amplification in Balanced Datasets Directional and Interpretable. arXiv:2412.11060, 2024.
> > >
> > > [5] Gupta, N., Smith, J., Adlam, B., and Mariet, Z. E. Ensembles of Classifiers: a Bias-Variance Perspective. TMLR, 2022.

---

### Official Review · Reviewer_5zXv · 2025-03-13

**Overall Recommendation:** 3

**Summary:**

The paper studies settings where predictions actively shape the data distribution, a formalism known as performative prediction. Beyond just predictive accuracy, their main focus is on understanding how different update procedures and learning algorithms influence the data distribution and understanding to what extent do performatively optimal predictors bias the distributions towards extreme values.

They study these questions in an outcome performative (predictions only influence outcomes not features) binary prediction setting. They further impose a linear model of how predictions shape the conditional distribution over outcomes. Their main results exactly characterize essentially all aspects of this binary problem: what is the performative optimal solution, what is the optimal loss, what is the bias between predictions and outcomes in the performatively optimal case, and how do predictions look like if one performs naive updates. They do this for both a stateless and stateful versions of their model.

**Claims And Evidence:**

Yes, all the claims in the submissions are supported by formal results.

**Essential References Not Discussed:**

I think that the paper would benefit by including a discussion to the following papers.

Kim & Perdomo (ITCS 2023). This paper studies a very similar setup where a learner optimizes an outcome performative binary outcome and designs learning algorithms that can both forecast outcomes accurately as well as steer them towards particular targets. (e.g. optimize the likelihood that y=1). Given the similarities in setup and focus on the broader impacts of prediction on the data beyond just accuracy, I think the paper would be improved by including it in the related work. Similarly, Miller et al ICML 2021 was one of the first papers to think about algorithms for performative optimality.

Outside of performative prediction, their dynamics model is nearly identical to the dynamics considered in Golowich et al Neurips 2024 in their paper on online control of population dynamics.

The focus of all of these papers is different to that considered here, but very much complementary. The reader would benefit from knowing how this work relates to the broader literature.

**Experimental Designs Or Analyses:**

The experimental designs are just visualizations of their theoretical results. These are valid and sound.

**Methods And Evaluation Criteria:**

Yes.

**Other Comments Or Suggestions:**

The estimators paragraph at the bottom of pg 5 has a typo “extend” ---> “extent”. Also it references Theorem 4.1 when it should be Proposition 4.1?

**Other Strengths And Weaknesses:**

The strengths of the paper is that it provides a lot of insight and a comprehensive analysis of the problem they study. The main weakness is that their setup is perhaps overly simplistic. They think about a scalar optimization problem over binary outcomes where the dynamics are highly constrained. It is not clear how these insights apply to other settings.

**Questions For Authors:**

NA

**Relation To Broader Scientific Literature:**

The paper contributes to the emerging area of performative prediction. Relative to other papers in this area, they focus on a very specific setup and aim to understand the long term impacts of predictions on the distribution instead of designing algorithms to exactly solve for optimal solutions.

Some of their main conceptual takeaways related to their finding that the performatively optimal solution is biased were known, at least in limited regimes. See for instance the biased coin toss example in the initial paper in the area (Perdomo et al 2020). However, they illustrate these ideas in a much richer setting.

**Theoretical Claims:**

I did not check the proofs in detail.

---

> ### Author Rebuttal · Authors · 2025-04-01
>
> Thank you for the thoughtful and constructive review! We address your concerns and questions below and remain available for further discussion.
>
> **I think that the paper would benefit by including a discussion to the following papers.**
>
> Thank you for the suggested references! We agree that, while orthogonal to our work, they are certainly relevant. We will cite and discuss them in the next version of our manuscript.
>
> Indeed, Kim and Perdomo (2023) investigate how to steer distributions towards a
> more desirable outcome by using omnipredictors. While steering is indeed
> different from accuracy maximization, the focus of this and our work are different. We
> analyze the secondary effects of performative risk minimization on the
> distribution and outcomes, which were not intended by the model provider. In contrast, Kim and Perdomo (2023) assume that the model provider
> intentionally changes the distribution according to their metric, which can be
> expressed in the omniprediction framework. Hence, the works could be seen a
> complementary. Our work allows the model provider to discover and analyze the
> unintended distribution changes, while the omniprediction could potentially
> allow the model provider to encode penalties for these changes, if they could be
> expressed in the omniprediction framework.
>
> Similarly, Golowich et al. (2024) focuses on distribution steering.
> However, their results do not inform the model provider about the unintended
> distribution changes due to the performative risk minimizaiton.
>
> As for Miller et al. (2021), we will make sure to add this paper in the existing
> discussion of learning algorithms in the performative setting.
>
> **The main weakness is that their setup is perhaps overly simplistic.**
>
> Our work indeed analyzes the specific learning problem of binary mean estimation (as clearly acknowledged in the title). Already on this classic learning problem, our work points towards potential shortcomings/side-effects of performative risk
> minimization. Therefore, while do not make specific claims for other settings, we hope that our work will encourage further exploration of the impact of performative methods.
>
> While the technical results are specific to the considered setup, we hope that our analysis can be a useful starting point for future exploration of other learning tasks for several reasons.
>
> First, while we only analyze the linear response model, we note that the
> linear response naturally arises as a first-order Taylor approximation for any
> performative response. Thus, we believe that our results will (at least
> qualitatively) transfer to **situations of weak performative response** (i.e.,
> small $|\alpha|$).
>
> Second, as we noted in the discussion of Lemma 3.2, **the error-uncertainty
> decomposition holds for a broad class of distributions**, not only the
> Bernoulli distribution. Thus, we can expect PRM to prefer distributions with
> less aleatoric uncertainty, which are usually more degenerate. For example, in
> the case of the multinomial distribution, the model provider would try to
> concentrate the probability mass on a small subset of outcomes.
>
> Third, our results can easily be extended to the following **more general group
> setting**. Imagine that clients consist of several independent groups, and each
> group reacts to the predictions of the model in the same way as the whole
> distribution in our paper. Also, assume that the model provider additionally
> gets some covariates that determine group membership before making a
> prediction. This modification makes our problem much closer to the usual
> supervised learning tasks where the model provider needs to simultaneously
> learn a model for membership prediction and outcomes for each group. At the
> same time, our results in the perfect information setting can be directly
> applied to this setup by independently applying the previous analysis to each
> group. The main limitation of such an extension is the assumption that groups
> evolve independently. This assumption could hold in the setting of drug
> efficacy prediction, but it will probably not hold in
> traffic prediction.
>
> **Typos**
>
> Thank you! We will fix these typos in the next version of our manuscript.

---

### Official Review · Reviewer_5pBv · 2025-03-13

**Overall Recommendation:** 4

**Summary:**

This paper studied the long-term impact of performative prediction on the predictor quality (as measured by bias) and population distribution (as measured by shift). It examined separately the settings of slow and rapid classifier updates with or without perfect information. The paper analyzed the long-term bias and shift in the context of binary mean estimation and compare and contrast them with the naive setting. The generalizability is implied with RL simulations.

## update after rebuttal
Thank the authors for providing detailed responses to my questions. I think the strength of this paper generally outweighs its weakness, so  I would consider raising the recommendation.

**Claims And Evidence:**

All claims are supported by either theoretical or experimental evidences.

**Essential References Not Discussed:**

Not recognized.

**Experimental Designs Or Analyses:**

RL simulations validate the theoretical findings with information restriction. The simulated curves converge to the theoretical prediction.

**Methods And Evaluation Criteria:**

The method makes sense and the theoretical results are instructional.
Evaluatoin metrics: bias and shifts make sense and the benchmark being the naive path (where the performativity is absent) is a reasonable baseline to study the impact of performativity.

**Other Comments Or Suggestions:**

1. Right column on page 5: the second to last line contains "Theorem 4.1", does it mean "Proposition 4.1"?
2. Last sentence of the caption of Figure 5: it seems both top and bottom rows correspond to the slow case.

**Other Strengths And Weaknesses:**

This paper focuses on a novel problem and is the few work looking at the long-term societal impact of performative prediction. Although the methodology may be too simplified to capture practical situations, the findings on the classifier bias and distribution shift in the long run are insightful.

**Questions For Authors:**

To complement the findings, would the case where the classifier is myopic, i.e., optimize instantaneous objectives at every step instead of solving for a long-term policy, be different from the model considered in the paper? If so, in what way the current model is more advantageous; and if not, why would you or the society prefer one instead of the other?

**Relation To Broader Scientific Literature:**

The work builds on performative prediction and long-term fairness, the combination of which is an emerging study in the literature. It contributes a quantifiable framework for PRM’s societal impact. The limitation lies in that focusing on binary variables restricts connections to broader performativity literature.

**Theoretical Claims:**

I checked correctness of proofs in section 4 and Theorem 5.1, where no issues were spotted.

---

> ### Author Rebuttal · Authors · 2025-04-01
>
> Thank you for the thoughtful and constructive review! We address your concerns and questions below and remain available for further discussion.
>
> **Setting $\lambda = 0$ seems too restrictive, no? ...**
>
> Thank you for your comment. Indeed, in Section 4.2.2 we had to make assumptions in order to ensure that we have one optimal value of the mean in the second period, $p_1$. First, we need to assume that the mean in the first period is also fixed, $p_0 = const$. Secondly, to make our setting estimable, we need to have some restrictions on either $\pi$ or $\lambda$, such as $\pi = 0$, $\pi=p_0$, or $\lambda = 0$ (otherwise, $p_0$ would not be distinguishable from $\pi$). Given that the purpose of experiments was to see how exploration interferes with performative risk minimization, we chose $\lambda = 0$ since this case is harder in terms of exploration compared to $\pi=0$ or $\pi=p_0$.
>
> Notice, however, that one can observe the effects of self-evolution in our experiments in Section 5.2. There we use $\lambda \neq 0$. When $T \to \infty$, the mean evolves over a long period of time and, for small values of $|\alpha|$, converges to an equilibrium value.
>
> Some trends we observe there as $T \to \infty$ are as follows:
>
> - The equilibrium mean, $p^*$, goes closer to the extreme values with an increase in $\alpha$, when $\alpha > 0$.
> - For the slow deployment case and $\alpha < 0$, $p^*$ goes closer to zero with an increase in $|\alpha|$.
> - For the rapid deployment case and $\alpha < 0$, for small values of $|\alpha|$ the mean also goes closer to zero. However, for big values of $|\alpha|$ it starts to oscillate.
>
> **Can you provide more details on how to obtain the optimal $\theta_t$ ...**
>
> Thank you for raising this point, we will clarify this in the next version of the manuscript. We find the value function, $V$, using the value iteration algorithm. Due to the Blackwell's contraction mapping theorem and the boundedness of the loss function, this process converges. Then, we solve the optimization problem for $\theta_t$ directly.
>
> **To complement the findings, would the case where the classifier is myopic ...**
>
> The naive path exactly serves this purpose. The naive path corresponds to a situation when the classifier is myopic, and the performativity is present, while the bias and mean shift compare the PRM-induced distribution path when the performativity is present with the distribution path when the performativity is absent.
>
> **If so, in what way the current model is more advantageous ...**
>
> Our results generally indicate that the naive (myopic) path induces less bias and shift in the positive feedback case. Thus, if one is concerned about the impact of performativity (see Section 6) and could sacrifice accuracy, the naive (myopic) classifier could be preferable to the PRM one. The negative feedback case is more complicated. When the performativity is weak ($|\alpha|$ is small), the naive path introduces a bigger mean shift. However, when the performativity is big, in the infinite horizon case, the naive path is more stable (e.g., it will not have such behavior as depicted by Figure 5, bottom row, fourth plot).
>
> **Typos**
>
> Thank you! We will fix these typos in the next version of our manuscript.

---

### Decision · Program_Chairs · 2025-05-01

**Decision:**

Accept (poster)

**Comment:**

This paper studies how performative risk minimization affects quality of prediction and distribution shift in a particular setting of interest. The paper establishes several theoretical results for several study cases and complements these with simulations. Reviewers were generally positive about the paper's contribution, and appreciated the authors' efforts to study the long-term impacts of performative learning beyond the narrow goal of risk minimization. At the same time, reviewers felt that the particular setting studied was perhaps too simplified – albeit natural for the questions considered, this made it difficult to see how results would generalize to more realistic and practical settings. Some reviewers also found the related work and the connections currently made in the paper require some additions and a revision. The authors' rebuttal was helpful in addressing some of the concerns and clarifying several points. The authors should take all point made in the reviews into account and integrate them into the paper.